# Parallel HIV-1 fitness landscapes shape viral dynamics in humans and macaques that develop broadly neutralizing antibodies

Kai S Shimagaki[1,2], Rebecca M Lynch[3], John P Barton[1,2]*

[1]Department of Computational and Systems Biology, University of Pittsburgh School of Medicine, Pittsburgh, United States; [2]Department of Physics and Astronomy, University of Pittsburgh, Pittsburgh, United States; [3]Department of Microbiology, Immunology and Tropical Medicine, School of Medicine and Health Sciences, George Washington University, Washington, DC, United States

## eLife Assessment

In this **important** quantitative study of HIV-1 evolution in humans and rhesus macaques, selection coefficients are inferred at scale over the HIV genome. Selection coefficients are similar in humans and macaques, providing **compelling** evidence that these coefficients are representative of the fitness landscapes of these viruses within hosts. This work will be of interest to the community working on quantitative evolution and fitness landscape inference, and the finding that rapid fitness gains in the HIV population predict bNAb emergence has significant implications for HIV vaccine design.

*For correspondence:
jpbarton@pitt.edu

## Abstract

HIV-1 evolves within individual hosts to escape adaptive immune responses while maintaining its capacity for replication. Coevolution between HIV-1 and the immune system generates extraordinary viral genetic diversity. In some individuals, this process also results in the development of broadly neutralizing antibodies (bnAbs) that can neutralize many viral variants, a key focus of HIV-1 vaccine design. However, a general understanding of the forces that shape virus-immune coevolution within and across hosts remains incomplete. Here, we performed a quantitative study of HIV-1 evolution in humans and rhesus macaques, including individuals who developed bnAbs. We observed strong selection early in infection for mutations affecting HIV-1 envelope glycosylation and escape from autologous strain-specific antibodies, followed by weaker selection for bnAb resistance. The inferred fitness effects of HIV-1 mutations in humans and macaques were remarkably similar. Moreover, we observed a striking pattern of rapid HIV-1 fitness gains that precedes the development of bnAbs. Our work highlights strong parallels between infection in rhesus macaques and humans, and it reveals a quantitative evolutionary signature of bnAb development.

## Introduction

HIV-1 rapidly mutates and proliferates in infected individuals. The immune system is a major driver of HIV-1 evolution, as the virus accumulates mutations to escape from host T cells and antibodies (*Wei et al., 2003*; *Allen et al., 2005*; *Li et al., 2007*). Due to the chronic nature of HIV-1 infection, coupled with high rates of mutation and replication, HIV-1 genetic diversity within and between infected

**eLife digest** Viruses are genetic particles composed of DNA or RNA, encased by a protective protein shell called the capsid. They cannot reproduce independently and must infect a host cell to replicate. Many viruses mutate rapidly, allowing them to adapt to and evade the immune responses of their hosts.

For example, HIV-1, the virus that causes AIDS, has a high mutation rate, resulting in the emergence of many distinct variants of the virus. Therefore, an effective vaccine needs to be able to stimulate a special type of antibody known as broadly neutralizing antibody (bnAb). These large defense proteins can recognize and neutralize many different viral strains, which could make them a key focus in HIV vaccine development.

Researchers often use rhesus macaques as a model system to study how HIV-1 evolves and interacts with the immune system. Previous studies have shown that some viruses mutate in similar ways in both humans and rhesus macaques. However, the details of HIV-1 evolution and mutation patterns in these two hosts remain unclear. Gaining deeper insight into the evolutionary processes linked to bnAb development could inform vaccine design and evaluate the suitability of rhesus macaques as an animal model for HIV-1 research.

Shimagaki et al. aimed to quantify how HIV-1 evolves in different hosts and whether these evolutionary patterns differ between individuals who do or do not develop bnAbs. The researchers reanalyzed previously collected HIV-1 data from two humans who developed bnAbs and 13 rhesus macaques, using computational models to estimate how various mutations affect viral replication (i.e., viral fitness). Their analysis revealed strong quantitative similarities in viral evolution between humans and macaques: the estimated fitness effects of mutations were highly correlated across species. Rapid increases in viral fitness were observed before bnAbs were detected, suggesting that selective pressure on the virus may help drive the development of antibody breadth.

These findings suggest that vaccine strategies designed to replicate the conditions that lead to rapid viral adaptation may help stimulate broadly neutralizing antibody responses. The observed parallels in HIV-1 evolution between humans and rhesus macaques also support the continued use of macaques as a relevant model for HIV-1 research. Still, significant challenges remain. Future studies should explore the link between viral evolution and antibody development in larger cohorts. Moreover, vaccine development requires addressing many practical aspects – such as antigen selection and dosing regimens – which extend beyond the viral fitness dynamics explored in this study.

individuals is incredibly high. Genetic diversity challenges vaccine development, as vaccine-elicited antibodies must be able to neutralize many strains of the virus to protect against infection (*Altfeld and Allen, 2006*).

However, there exist rare antibodies that are capable of neutralizing a broad range of HIV-1 viruses. These broadly neutralizing antibodies (bnAbs) have therefore been the subject of intense research (*Kwong et al., 2013*; *Burton and Hangartner, 2016*; *Sok and Burton, 2018*; *Haynes et al., 2023*). Eliciting bnAbs through vaccination remains a major goal of HIV-1 vaccine design. However, the development of exceptionally broad antibody responses is rare, and such antibodies typically develop only after several years of infection (*Doria-Rose et al., 2010*; *Haynes et al., 2012*; *Haynes et al., 2016*; *Hraber et al., 2014*).

Recent years have yielded important insights into the coevolutionary process between HIV-1 and antibodies that sometimes leads to the development of bnAbs. Clinical studies have collected serial samples of HIV-1 sequences from a few individuals who developed bnAbs and characterized the resulting antibodies, their developmental stages, and binding sites (*Liao et al., 2013*; *Bonsignori et al., 2017*; *Doria-Rose et al., 2014*). The contributions of HIV-1 and its coevolution to bnAb development are complex (*Moore et al., 2015*; *Landais and Moore, 2018*). High viral loads and viral diversity have been positively associated with bnAb development (*Gray et al., 2011*; *Moore et al., 2015*; *Landais and Moore, 2018*). However, superinfection, which can vastly increase HIV-1 diversity, is not always associated with bnAb development (*Cornelissen et al., 2016*), and it does not appear to broaden antibody responses in the absence of other factors (*Landais and Moore, 2018*).

Here, we sought to characterize the evolutionary dynamics of HIV-1 that accompany the development of bnAbs in clinical data. In particular, we inferred the landscape of selective pressures that shape the evolution of HIV-1 within hosts, reflecting the effects of the immune environment. We first analyzed data from two individuals who developed bnAbs within a few years after HIV-1 infection (*Liao et al., 2013*; *Bonsignori et al., 2017*). In both individuals, HIV-1 mutations inferred to be the most beneficial were observed early in infection. In general, mutations that provided resistance to autologous strain-specific antibodies were inferred to be more strongly selected than ones that escaped from bnAbs. We also observed clusters of beneficial mutations along the HIV-1 genome, which were associated with envelope protein (Env) structure.

To confirm the generality of these patterns in a broader sample, we studied recent data from rhesus macaques (RMs) infected with simian-human immunodeficiency viruses (SHIV) that incorporated HIV-1 Env proteins derived from the two individuals above (*Roark et al., 2021*). This study also compared patterns of Env evolution in HIV-1 and SHIV in response to host immunity. We observed striking parallels between the inferred fitness effects of Env mutations in RMs and humans, suggesting highly similar selective pressures on the virus despite different host species and differences in individual immune responses. Furthermore, we found that RMs that developed broad, potent antibody responses could clearly be distinguished from those with narrowly focused responses using the evolutionary dynamics of the virus. Specifically, the virus population in individuals who developed greater breadth was distinguished by larger and more rapid gains in fitness than in other individuals. Collectively, these results show high similarity between SHIV evolutionary dynamics in RMs and HIV-1 in humans, and that viral fitness gain is associated with antibody breadth.

## Results

### Quantifying HIV-1 evolutionary dynamics

We studied HIV-1 evolution accompanying the development of bnAbs in two donors, CH505 and CH848, enrolled in the Center for HIV/AIDS Vaccine Immunology 001 acute infection cohort (*Tomaras et al., 2008*). CH505 developed the CD4 binding site-targeting bnAb CH103, which was first detectable 14 weeks after HIV-1 infection (*Liao et al., 2013*). CH103 maturation was found to be associated with viral escape from another antibody lineage, CH235, that ultimately developed significant breadth (*Gao et al., 2014*; *Bonsignori et al., 2016*). CH848 developed a bnAb, DH270, targeting a glycosylated site near the third variable loop (V3) of Env (*Bonsignori et al., 2017*). Similar to the bnAb development process in CH505, escape from 'cooperating' DH272 and DH475 lineage antibodies was observed to contribute to the maturation of DH270 (*Bonsignori et al., 2017*).

To quantify HIV-1 evolutionary dynamics, we sought to infer a fitness model that best explained the changes in the genetic composition of the viral population observed in each individual over time. In recent years, a wide variety of approaches have been developed to infer the fitness effects of mutations from temporal genetic data (*Bollback et al., 2008*; *Illingworth and Mustonen, 2011*; *Illingworth and Mustonen, 2012*; *Malaspinas et al., 2012*; *Mathieson and McVean, 2013*; *Lacerda and Seoighe, 2014*; *Feder et al., 2014*; *Steinrücken et al., 2014*; *Foll et al., 2014*; *Terhorst et al., 2015*; *Schraiber et al., 2016*; *Tataru et al., 2017*; *Paris et al., 2019*; *Mathieson and Terhorst, 2022*; *He et al., 2023*; *Sohail et al., 2022*; *Sohail et al., 2021*; *Shimagaki and Barton, 2025b*; *Gao and Barton, 2025*; *Lee et al., 2025*). The vast majority of these methods focus on a single locus at a time, ignoring correlations between genetic variants at different loci. While the recombination rate of HIV-1 is high (*Neher and Leitner, 2010*; *Romero and Feder, 2024*), the virus also evolves under strong selection, which can lead to interference between clones with different beneficial mutations (*Rouzine and Weinberger, 2013*; *Pandit and de Boer, 2014*; *Garcia and Regoes, 2014*; *Garcia et al., 2016*; *Williams and Pennings, 2020*; *Sohail et al., 2021*). Thus, we applied MPL, an inference method that systematically accounts for genetic correlations (*Sohail et al., 2022*; *Sohail et al., 2021*; *Shimagaki and Barton, 2025b*; *Gao and Barton, 2025*; *Lee et al., 2025*), to estimate the fitness effects of HIV-1 mutations.

### Model overview

Here, we provide a brief overview of the key steps in the MPL approach to inferring selection. Further details are available in Methods and in prior work (*Sohail et al., 2022*; *Sohail et al., 2021*; *Shimagaki*

and Barton, 2025b; Gao and Barton, 2025; Lee et al., 2025). First, we assume that the effect on viral fitness of each individual mutation $a$ at each site $i$ is quantified by a selection coefficient $s_i(a)$, with positive coefficients $s_i(a) > 0$ denoting mutations that are beneficial for the virus and $s_i(a) < 0$ denoting deleterious ones. We further assume that the cumulative fitness effects of mutations are additive, such that the overall fitness $F^\alpha$ of a viral sequence α is given by the sum of the selection coefficients for all the mutations that it bears. That is,

$$F^\alpha = F(g^\alpha) = 1 + \sum_{i=1}^{L} \sum_{a=1}^{q} s_i(a) g_{i,a}^\alpha,$$ (1)

where $g^\alpha = ((g_{i,a}^\alpha)_{a=1}^{q})_{i=1}^{L}$ represents the viral sequence, with $g_{i,a}^\alpha$ equal to one if genotype α has allele $a$ at site $i$ and zero otherwise. $L$ is the length of the genetic sequence, and $q$ represents the number of genetic states (i.e., $q = 5$ for nucleotide sequences and 21 for amino acids, including gaps/deletions).

We assume that viral replication is stochastic, where viruses with higher fitness are more likely to spread infection to new cells than ones with lower fitness. Let us write the number of viruses of each genotype in the population at time $t$ as $n(t) = (n_1(t), n_2(t), \ldots, n_M(t))$, where $M = q^L$ is the total number of possible genotypes. In our model, the probability of obtaining a new distribution of genotypes $n(t+1)$ in the next generation is multinomial,

$$P\left(n(t+1)\right) = N! \prod_{\alpha=1}^{M} \frac{p_\alpha(n(t))^{n_\alpha(t+1)}}{n_\alpha(t+1)!},$$ (2)

with $N = \sum_\alpha n_\alpha$ the total population size. The probabilities $p_\alpha(n(t))$ are influenced by fitness, as well as mutation, recombination, and the current frequency of each genotype (see Methods). Essentially, viruses that are fitter than the current population average are more likely to increase in frequency, while ones that are less fit are likely to decline. Mutations and recombination introduce genetic variation into the population. The population size $N$ determines the relative stochasticity of the dynamics, with smaller populations having more random fluctuations than larger ones.

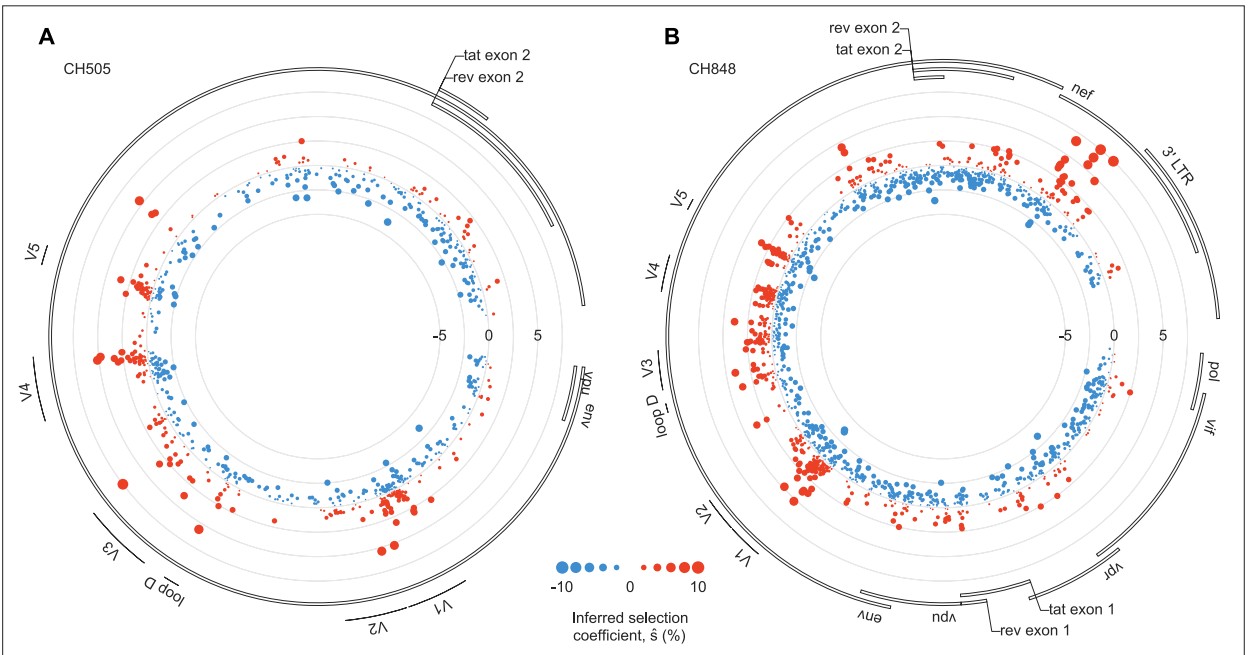

**Figure 1.** Beneficial mutations occur in clusters along the genome. Inferred fitness effects of HIV-1 mutations in CH505 (**A**) and CH848 (**B**). The position along the radius of each circle specifies the strength of selection: mutations plotted closer to the center are more deleterious, while those closer to the edge are more beneficial. For both individuals, clusters of beneficial mutations are observed in the variable loops of Env, some of which are associated with antibody escape. For CH848, a group of strongly beneficial mutations also appears in Nef.

The online version of this article includes the following figure supplement(s) for figure 1:

**Figure supplement 1.** Weak correlation between sequence variability, as measured by entropy, and inferred selection.

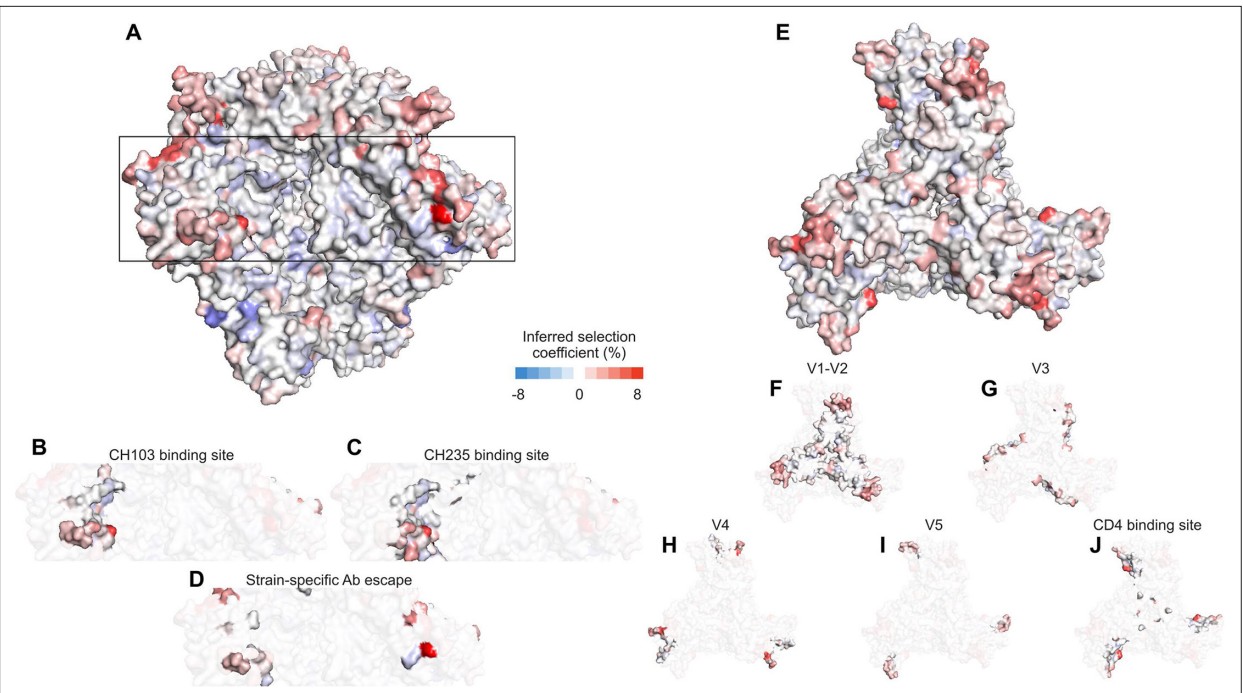

**Figure 2.** Visualization of the effects of HIV-1 mutations in CH505 on the Env trimer. (**A**) Side view of the Env trimer (*Jardine et al., 2016*), with detail views of selection for mutations in the CH103 binding site (**B**) CH235 binding site (**C**) and sites associated with escape from autologous strain-specific antibodies (**D**). (**E**) Top view of the trimer, with detail views of variable loops (**F-I**) and the CD4 binding site (**J**). Generally, beneficial mutations appear more frequently near exposed regions at the top of the Env trimer, and deleterious mutations appear in more protected regions. Mutations near the V1/V2 apex can affect the binding and neutralization of antibodies targeting the CD4 binding site (*Gao et al., 2014*).

The online version of this article includes the following figure supplement(s) for figure 2:

**Figure supplement 1.** CH848 exhibits similar spatial patterns of selection coefficients.

Following this model, we can then quantify the probability of any evolutionary history of the viral population (i.e., the distribution of viral genotypes over time) as a function of the selection coefficients (Methods).

Assuming that the population size $N$ is large and the selection coefficients are small (such that $|s| \ll 1$), we can write an analytical expression for the selection coefficients that best fit the viral dynamics observed in data (*Sohail et al., 2022*; *Sohail et al., 2021*):

$$\hat{s} = \left( C_{\text{int}} + \gamma I \right)^{-1} \left[ \Delta x_{\text{int}} - \Delta u_{\text{int}} \right] . \tag{3}$$

Here, $C_{\text{int}}$ is the covariance matrix of allele frequencies (i.e. the linkage disequilibrium matrix) integrated over time, and $\gamma$ is a regularization parameter. The terms $\Delta x_{\text{int}}$ and $\Delta u_{\text{int}}$ represent the net change in allele frequency and the total expected change in allele frequency due to spontaneous mutations alone, respectively (see Methods for details). Intuitively, this expression says that net allele frequency changes that cannot be explained by mutations are likely due to selection, either on that specific allele or the associated genetic background, which is quantified by $C_{\text{int}}$. Alleles that have large, rapid changes in frequency are more likely to be under strong selection than those with smaller, slower frequency changes. Beyond HIV-1 (*Sohail et al., 2021*), this approach has been successfully applied to study the evolution of SARS-CoV-2 (*Lee et al., 2025*) and experimental evolution in bacteria (*Li and Barton, 2023*; *Li and Barton, 2024*).

## Broad patterns of HIV-1 selection

As described above, we used MPL (*Sohail et al., 2022*; *Sohail et al., 2021*) to infer the selection coefficients that best fit the viral dynamics observed in data from CH505 and CH848. While the great majority of HIV-1 mutations were inferred to be neutral ($s_i(a) \sim 0$), a few mutations substantially

increase viral fitness (**Figure 1**). Strongly beneficial mutations occurred in clusters along the genome and preferentially appeared in specific regions of Env (**Figure 2**).

To quantify patterns of selection in Env, we examined the top 2% of mutations inferred to be the most beneficial in CH505 and CH848. The fractions of nonsynonymous mutations within these subsets were 97% and 92% for CH505 and CH848, respectively. These fractions are significantly higher than chance expectations ($p = 6.7 \times 10^{-3}$ and $5.6 \times 10^{-4}$, Fisher's exact test; Methods), supporting the model's ability to accurately infer fitness effects in this data. For CH505, we found 10.9-fold more strongly beneficial mutations in the first variable loop (V1) than expected by chance ($p = 2.5 \times 10^{-3}$). This is consistent with the presence of V1 mutations conferring resistance to autologous strain-specific antibodies (**Gao et al., 2014**). Mutations in V4, a region targeted by CD8 + T cells (**Gao et al., 2014**), were also 10.0-fold enriched in this subset ($p = 9.0 \times 10^{-5}$). For CH848, mutations in V1, V3, and V5 were enriched by factors of 14.2, 6.3, and 19.1 among the top 2% most beneficial mutations ($p = 5.9 \times 10^{-10}$, $6.6 \times 10^{-3}$, and $4.8 \times 10^{-6}$). Mutations in these regions were shown to play a role in resistance to DH270 and DH475 lineage antibodies (**Bonsignori et al., 2017**). To test whether our results might be biased by overall sequence variability, we examined the relationship between our inferred selection coefficients and entropy, a common measure of sequence variability. Overall, we found only a modest correlation between selection and entropy, suggesting that the signs of selection that we observe are not due to increased sequence variability alone (**Figure 1—figure supplement 1**).

Reversions and mutations affecting N-linked glycosylation motifs were also likely to be beneficial. We define reversions as mutations where the transmitted/founder (TF) amino acid changes to match the subtype consensus sequence at the same site. Among the top 2% most beneficial mutations, reversions were enriched by factors of 19.9 and 17.8 for CH505 and CH848 viruses, respectively ($p = 2.1 \times 10^{-8}$ and $8.5 \times 10^{-13}$), consistent with past work finding strong selection for reversions (**Zanini et al., 2015**; **Sohail et al., 2021**). For CH848, this group also includes several strongly selected mutations observed in Nef (**Figure 1B**), a protein that plays multiple roles during HIV-1 infection (**Das and Jameel, 2005**; **Barton et al., 2019**). Mutations affecting N-linked glycosylation motifs (i.e., by adding, removing, or shifting a glycosylation motif) were enriched by factors of 4.6 ($p = 7.0 \times 10^{-3}$)

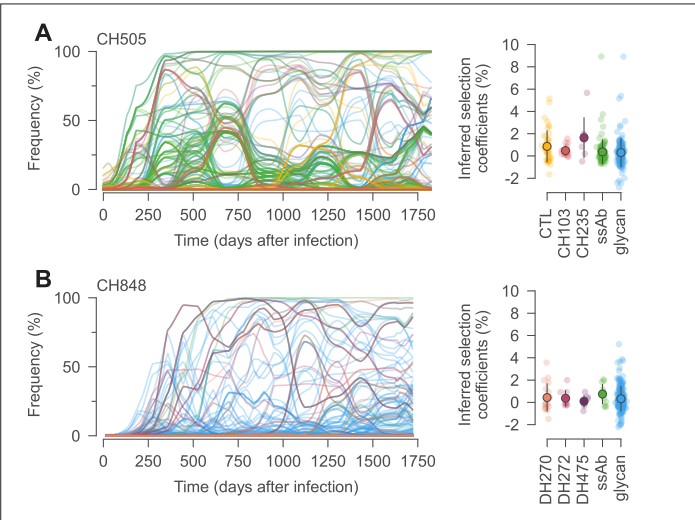

**Figure 3.** Trajectories and inferred selection coefficients for immune escape mutations and mutations affecting Env glycosylation. (**A**) For CH505, early, strongly selected mutations include ones that escape from CD8 + T cells and autologous strain-specific antibodies (ssAbs). More moderately selected bnAb resistance mutations tend to arise later. Note that mutations that affect bnAb resistance can appear in the viral population before bnAbs are generated. Open circles and error bars reflect the mean and standard deviation of inferred selection coefficients in each category. (**B**) For CH848, mutations affecting glycosylation dominate the early phase of evolution, followed later by mutations affecting bnAb resistance. For easier visualization, frequency trajectories are shown with exponential smoothing with a time scale of $t = 50$ days. See **Figure 2**, **Figure 2—figure supplement 1** for a detailed view of mutation frequency trajectories by mutation type without smoothing.

The online version of this article includes the following figure supplement(s) for figure 3:

**Figure supplement 1.** Frequencies of different types of HIV-1 mutations over time.

and 8.7 ($p = 1.4 \times 10^{-10}$). Changes in glycosylation patterns contributed to antibody escape for both CH505 and CH848 (*Gao et al., 2014*; *Bonsignori et al., 2017*).

## Selection for antibody escape

To quantify levels of selection for antibody escape, we computed selection coefficients for mutations that were observed to contribute to resistance to bnAbs as well as bnAb precursors and autologous strain-specific antibodies (*Supplementary file 1* and *Supplementary file 3*; *Gao et al., 2014*; *McCurley et al., 2017*; *Kong et al., 2019*; *Saunders et al., 2022*; *Bauer et al., 2023*). We mapped the inferred selection coefficients to the Env protein structure (*Liao et al., 2013*), highlighting the binding sites for bnAbs and resistance mutations for strain-specific antibodies, as well as important parts of Env (*Figure 2*, *Figure 1—figure supplement 1*). We also analyzed when these resistance mutations were first observed in each individual.

Overall, we observed stronger selection for escape from autologous strain-specific antibodies and/or changes in glycosylation during the first six months of infection (*Supplementary file 4* and *Supplementary file 5*). This was then followed by more modest selection for escape from bnAb lineages (*Figure 3*). For CH505, mutations that conferred resistance to the intermediate-breadth CH235 lineage (*Gao et al., 2014*) were less beneficial than top mutations escaping from strain-specific antibodies (*Figure 3A*). In turn, resistance mutations for the broader CH103 lineage were less beneficial than CH235 resistance mutations (*Figure 3A*). CH848 is similar, with some highly beneficial mutations affecting glycosylation observed early in infection (*Figure 3B*). While a few mutations affecting DH270 appear strongly selected, these mutations appeared long before the DH270 lineage was detected (around 3.5 years after infection *Bonsignori et al., 2017*). Thus, these mutations may have initially been selected for other reasons.

## Consistent patterns of selection in SHIV evolution

Simian-human immunodeficiency viruses (SHIVs) have numerous applications in HIV/AIDS research (*Hatziioannou and Evans, 2012*; *Li et al., 2016*). Recently, Roark and collaborators studied

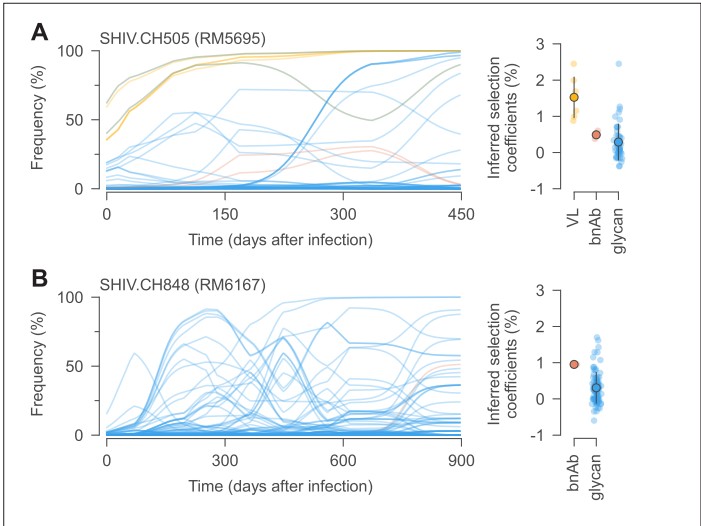

**Figure 4.** Example trajectories and selection coefficients for SHIV mutations that affect viral load, bnAb recognition, or glycosylation. (**A**) In RM5695, infected with SHIV.CH505, mutations known to increase viral load (*Bauer et al., 2023*) and ones affecting glycosylation were rapidly selected. Mutations affecting resistance to broad antibodies (*Roark et al., 2021*) arose later under moderate selection. Open circles and error bars reflect the mean and standard deviation of inferred selection coefficients in each category. (**B**) Slower but qualitatively similar evolutionary patterns were observed in RM6163, infected with SHIV.CH848. For easier visualization, frequency trajectories are shown with exponential smoothing with a time scale of $t = 50$ days.

The online version of this article includes the following figure supplement(s) for figure 4:

**Figure supplement 1.** The number of RMs in which a SHIV mutation was observed is only weakly associated with its inferred fitness effect.

SHIV-antibody coevolution in rhesus macaques (RMs), which they compared with patterns of HIV-1 evolution (*Roark et al., 2021*). Two of the SHIV constructs in this study included envelope sequences derived from CH505 and CH848 transmitted/founder (TF) viruses. There, it was found that 2 out of 10 RMs inoculated with SHIV.CH505 and 2 out of 6 RMs inoculated with SHIV.CH848 developed antibodies with substantial breadth.

To understand whether the patterns of HIV-1 selection observed in CH505 and CH848 are repeatable, and to search for viral factors that distinguish between individuals who develop bnAbs and those who do not, we analyzed SHIV.CH505 and SHIV.CH848 evolution in RMs (*Roark et al., 2021*). To prevent spurious inferences, we first omitted data from RMs with <3 sampling times or <4 sequences in total (Methods). After processing, we examined evolutionary data from seven RMs inoculated with SHIV.CH505 and six RMs inoculated with SHIV.CH848 (*Supplementary file 6* and *Supplementary file 7*). We then computed selection coefficients for SHIV mutations within each RM. Reasoning that selective pressures across SHIV.CH505 and SHIV.CH848 viruses are likely to be similar, we also inferred two sets of joint selection coefficients that best describe SHIV evolution in SHIV.CH505- and SHIV.CH848-inoculated RMs, respectively (Methods).

As before, we examined the top 2% of SHIV.CH505 and SHIV.CH848 mutations that we inferred to be the most beneficial for the virus. Overall, we found consistent selection for reversions (17.9- and 14.2-fold enrichment, $p = 4.3 \times 10^{-11}$ and $1.2 \times 10^{-11}$ for SHIV.CH505 and SHIV.CH848, respectively), with slightly attenuated enrichment in mutations that affect N-linked glycosylation (2.7- and 4.2-fold enrichment, $p = 5.4 \times 10^{-2}$ and $3.4 \times 10^{-6}$). However, there is a small subset of mutations that shift glycosylation sites by simultaneously disrupting one N-linked glycosylation motif and completing another, where highly beneficial mutations occur far more often than expected by chance (158.3- and 191.1-fold enrichment, $p = 1.7 \times 10^{-4}$ and $7.9 \times 10^{-11}$ for CH505 and SHIV.CH505, respectively; 118.7-fold and 90.4-fold enrichment, $p = 2.0 \times 10^{-4}$ and $2.3 \times 10^{-7}$ for CH848 and SHIV.CH848).

Intuitively, one may expect that strongly beneficial SHIV mutations are more likely to be observed in samples from multiple RMs. However, we found that the number of RMs in which a mutation is observed is only weakly associated with the fitness effect of the mutation (*Figure 4*, *Figure 4—figure supplement 1*). While substantially deleterious SHIV mutations are rarely observed across multiple RMs, neutral and nearly neutral mutations are common. Thus, in this data set, it is not generally true that SHIV mutations observed in multiple hosts must significantly increase viral fitness.

We observed some differences between HIV-1 and SHIV in the precise locations of the most beneficial Env mutations. For example, mutations in V4 are highly enriched in CH505 due to a CD8 + T cell epitope in this region, but not in SHIV.CH505 (2.7-fold enrichment, $p = 0.13$). For SHIV.CH848, beneficial mutations are modestly enriched in V1 and V5 (4.7- and 4.2-fold, $p = 4.0 \times 10^{-3}$ and $6.7 \times 10^{-2}$), as in CH848, but not for V3 (0.9-fold enrichment, $p = 0.22$).

Despite some differences in the top mutations, patterns of selection over time in SHIV were very similar to those found for HIV-1. As before, highly beneficial mutations, including ones affecting glycosylation, tended to appear earlier in infection. This was followed by modestly beneficial mutations at later times, including ones involved in resistance to bnAbs in the RMs who developed antibodies with significant breadth (examples in *Figure 4*).

## Detection of SHIV mutations that increase viral load

A major goal of nonhuman primate studies with SHIV is to faithfully recover important aspects of HIV-1 infection in humans. However, due to the divergence of simian immunodeficiency viruses and HIV-1, SHIVs are not always well-adapted to replication in RMs (*Bauer et al., 2023*). To combat this problem, a recent study identified six SHIV.CH505 mutations that increase viral load (VL) in RMs (*Bauer et al., 2023*). These mutations result in viral kinetics that better mimic HIV-1 infection in humans.

Our analysis readily identifies the SHIV.CH505 mutations shown to increase VL. Five out of the top six SHIV.CH505 mutations with the largest average selection coefficients are associated with increased VL (*Supplementary file 8*). The final mutation identified by Bauer et al., N130D, is ranked tenth. We also find highly beneficial mutations in SHIV.CH848 that are distinct from those in SHIV.CH505 (*Supplementary file 9*). Highly ranked mutations identified here may be good experimental targets for future studies aimed at increasing SHIV.CH848 replication in vivo.

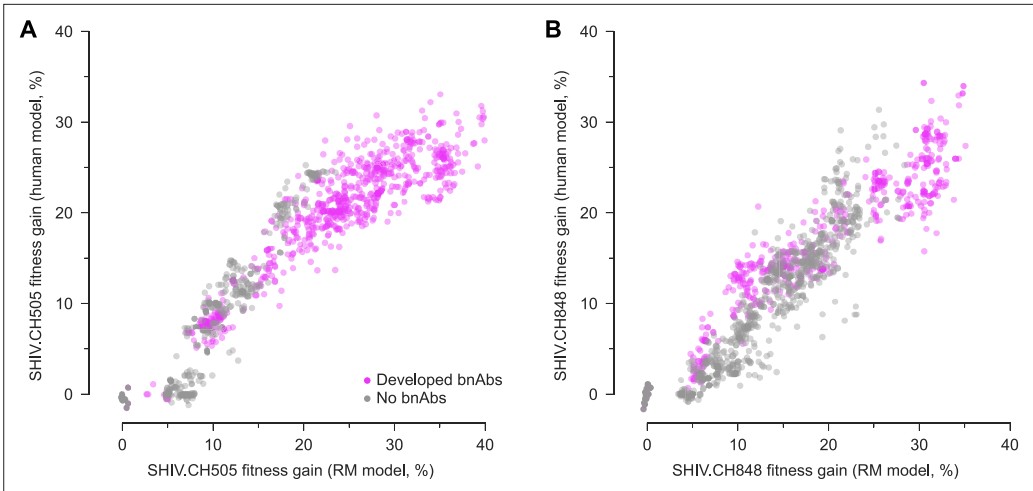

**Figure 5.** Inferred fitness landscapes in HIV-1 and SHIV are highly similar. (**A**) Fitness of SHIV.CH505 sequences relative to the TF sequence across 7 RMs, including 2 that developed bnAbs, 1 that developed tier 2 nAbs that lacked a critical mutation for breadth, and 4 that did not develop broad antibody responses, evaluated using fitness effects of mutations using data from CH505 and using RM data. The fitness values are strongly correlated, indicating the similarity of Env fitness landscapes inferred using HIV-1 or SHIV data. Values are normalized such that the fitness gain of the TF sequence is zero. (**B**) Fitness values for SHIV.CH848 sequences also show strong agreement between CH848 and SHIV.CH848 landscapes.

The online version of this article includes the following figure supplement(s) for figure 5:

**Figure supplement 1.** Randomized sequences show broad similarity between HIV-1 and SHIV fitness landscapes.

**Figure supplement 2.** Little correlation in fitness values estimated from evolutionarily distant sequences.

## Fitness agreement between HIV-1 and SHIV

Next, we explored the similarity in the overall viral fitness landscapes inferred for HIV-1 and SHIV, beyond just the top mutations. First, we computed the fitness of each SHIV sequence using the joint SHIV.CH505 and SHIV.CH848 selection coefficients inferred from RM data. Then, we computed fitness values for SHIV sequences using selection coefficients inferred from HIV-1 evolution in CH505 and CH848.

We observed a remarkable agreement between SHIV fitness values computed from these two sources (*Figure 5*). For both SHIV.CH505 and SHIV.CH848, the correlation between viral fitness estimated using data from humans (i.e. CH505 and CH848) and RMs is strongly and linearly correlated (Pearson's $r = 0.96$ and $0.95$, $p < 10^{-20}$). This implies that evolutionary pressures on the envelope protein SHIV-infected RMs are highly similar to those on HIV-1 in humans with the same TF Env. In fact, this relationship holds even beyond the SHIV sequences observed during infection. Fitness estimates for sequences with randomly shuffled sets of mutations are also strongly correlated (*Figure 5—figure supplement 1*). In contrast, the inferred fitness landscapes of CH505 and CH848, which share few mutations in common, are poorly correlated (*Figure 5—figure supplement 2*). This suggests that the similarities between viral fitness values in humans and RMs are not artifacts of the model, but rather stem from similarities in underlying evolutionary drivers.

## Evolutionary dynamics forecast antibody breadth

Given the similarity of HIV-1 and SHIV evolution, we sought to identify evolutionary features that distinguish between hosts who develop broad antibody responses and those who do not. *Figure 5* shows that SHIV sequences from hosts with bnAbs often reach higher fitness values than those in hosts with only narrow-spectrum antibodies. We hypothesized that stronger selective pressures on the virus might drive viral diversification, stimulating the development of antibody breadth. Past studies have associated higher viral loads with bnAb development and observed viral diversification around the time of bnAb emergence (*Liao et al., 2013*; *Moore et al., 2015*; *Landais and Moore, 2018*). Computational studies and experiments have also shown that sequential exposure to diverse antigens can induce cross-reactive antibodies (*Wang et al., 2015*; *Escolano et al., 2016*; *Sprenger et al., 2020*).

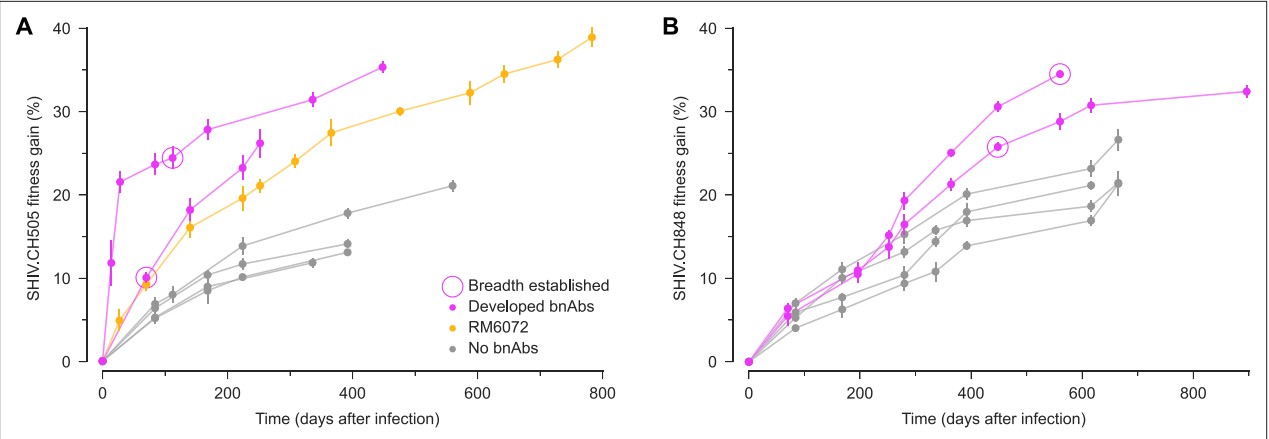

**Figure 6.** Rapid SHIV fitness gains precede the development of broadly neutralizing antibodies. For both SHIV.CH505 (**A**) and SHIV.CH848 (**B**), viral fitness gains over time display distinct patterns in RM hosts that developed bnAbs versus those that did not. Notably, the differences in SHIV fitness gains between hosts with and without broad antibody responses appear before the development of antibody breadth and cannot be attributed to selection for bnAb resistance mutations. RM6072 is an unusual case, exhibiting antibody development that was highly similar to CH505. Although RM6072 developed tier 2 nAbs, they lacked key mutations critical for breadth (***Roark et al., 2021***). Points and error bars show the mean and standard deviation of fitness gains across SHIV samples in each RM at each time.

The online version of this article includes the following figure supplement(s) for figure 6:

**Figure supplement 1.** Contributions of different types of SHIV mutations to viral fitness gains over time.

**Figure supplement 2.** Distribution of inferred epistatic interactions.

**Figure supplement 3.** Similarity between effective selection coefficients obtained from the epistatic model and selection coefficients in the additive model.

**Figure supplement 4.** Comparison between fitness values in the additive and epistatic models.

**Figure supplement 5.** Robustness of the inferred selection coefficients using bootstrap resampling.

To further quantify SHIV evolutionary dynamics, we computed the average fitness gain of viral populations in each RM over time. We observed a striking difference in SHIV fitness gains between RMs that developed broad antibody responses and those that did not (***Figure 6***). In particular, SHIV fitness increased rapidly before the development of antibody breadth. SHIV fitness gains in RM5695, which developed exceptionally broad and potent antibodies (***Roark et al., 2021***), were especially rapid and dramatic. These fitness differences were not attributable to bnAb resistance mutations, which were only moderately selected and generally appeared after bnAbs developed.

One outlier in this pattern is RM6072, infected with SHIV.CH505. Antibody development in RM6072 followed a path that was remarkably similar to CH505, including a lineage of antibodies, DH650, directed toward the CD4 binding site of Env (***Roark et al., 2021***). However, resistance to the DH650 lineage is conferred by a strongly selected mutation that adds a glycan at site 234 (T234N, with an inferred selection coefficient of 4.5%). Broadly neutralizing antibodies similar to DH650 are able to accommodate this glycan due to shorter and/or more flexible light chains (***Zhou et al., 2013***; ***Roark et al., 2021***), but DH650 cannot. Antibody evolution in RM6072 thus proceeded along a clear pathway toward bnAb development but lacked critical mutations to achieve breadth.

Next, we quantified how different types of SHIV mutations contributed to viral fitness gains over time. We examined contributions from VL-enhancing mutations (***Bauer et al., 2023***), antibody escape mutations (***Bauer et al., 2023***; ***Roark et al., 2021***), other mutations affecting Env glycosylation, and reversions to subtype consensus. We found increased fitness gains across all types of mutations in RMs that developed broad antibody responses, compared to those that did not (***Figure 6—figure supplement 1***). VL-enhancing mutations, known antibody resistance mutations, and reversions typically made the largest contributions to viral fitness.

## Robustness of inferred selection to changes in the fitness model and finite sampling

In the analysis above, we used a simple model where the net fitness effect of multiple mutations is simply equal to the sum of their individual effects. Recently, methods have also been developed that can infer epistatic fitness effects from data, which include pairwise interactions between mutations (*Sohail et al., 2022*; *Shimagaki and Barton, 2025b*). We reanalyzed these data to examine how inferred fitness changes when epistasis is included in the model, using the approach of Shimagaki and Barton (2025; Methods). Overall, the inferred epistatic interactions were modest (*Figure 6—figure supplement 2*). In CH505, we found that the CD4 binding site, V1 (especially sites 136–146 in HXB2 numbering) and V5 regions were modestly but significantly enriched in the most beneficial (top 1%) of epistatic interactions (2.5-, 1.2-, and 1.8-fold enrichment with $p = 1.0 \times 10^{-21}, 6.3 \times 10^{-6}$ and $6.3 \times 10^{-5}$, respectively). Epistatic interactions between N280S/V281A and E275K/V281G, which confer resistance to CH235 (*Gao et al., 2014*), ranked in the top 6.5% and 13.0% of interactions. In CH848, we found 1.3-, 1.5-, and 2.3-fold enrichment in strong beneficial epistatic interactions in the CD4 binding site, V4, and V5 regions, respectively ($p = 4.0 \times 10^{-6}, p = 2.5 \times 10^{-14}$ and $3.2 \times 10^{-19}$).

To compare the typical fitness effects of individual mutations in the model with epistasis to those in the additive model, we computed effective selection coefficients for the epistatic model. For each mutant allele $a$ at each site $i$, we computed the average difference in fitness between sequences in the data set with the mutation and hypothetical sequences that are the same as those in the data, except with the mutant allele $a$ reverted to the TF one. In this way, the effective selection coefficient measures the typical effect of each mutation in the data set, while also accounting for epistatic interactions with the sequence background. We found that the effective selection coefficients were highly correlated with the selection coefficients from the additive model (*Figure 6—figure supplement 3*). We also found strong agreement between the additive and epistatic model fitness values for each sequence in both HIV-1 and SHIV data (*Figure 6—figure supplement 4*).

Finite sampling of sequence data could also affect our analyses. To further test the robustness of our results, we inferred selection coefficients using bootstrap resampling, where we resample sequences from the original ensemble, maintaining the same number of sequences for each time point and subject. The selection coefficients from the bootstrap samples are consistent with the original data (see *Figure 6—figure supplement 5* for a typical example), with Pearson's $r$ values of around 0.85 for HIV-1 data sets and 0.95 for SHIV data sets, respectively.

## Discussion

HIV-1 evolves under complex selective pressures within individual hosts, balancing replicative efficiency with immune evasion. Here, we quantitatively studied the evolution of HIV-1 and SHIV (featuring HIV-1-derived Env sequences) across multiple hosts, including some who developed broad antibody responses against the virus. Our study highlighted how different classes of mutations (e.g. mutations affecting T cell escape or Env glycosylation) affect fitness in vivo. In both HIV-1 and SHIV, we found strong selection for reversions to subtype consensus and some mutations that affected N-linked glycosylation motifs or resistance to autologous strain-specific antibodies. Few CD8 + T cell epitopes were identified in this data set, but the T cell escape mutations that we did observe were highly beneficial for the virus. Consistent with past work studying VRC26 escape in CAP256, we observed more modest selection for bnAb resistance mutations (*Sohail et al., 2021*).

Overall, we found striking similarities between Env evolution in humans and RMs. Importantly, these parallels extend beyond the observation of repetitive mutations: the number of hosts in which a mutation was observed was only weakly associated with the mutation's fitness effect (*Figure 4—figure supplement 1*). Our inferred Env fitness values in humans and RMs were highly correlated, indicating that the functional and immune constraints shaping Env evolution in HIV-1 and SHIV infection are very similar. Our findings, therefore, reinforce SHIV as a model system that closely mirrors HIV-1 infection.

We discovered that the speed of SHIV fitness gains was clearly higher in RMs that developed broad antibody responses than in those with narrow-spectrum antibodies. Fitness gains in the viral population preceded the development of bnAbs, and they were not driven by bnAb resistance mutations. This suggests that rapid changes in the viral population are a cause rather than a consequence of antibody breadth. While our sample is limited to 13 RMs and two founder Env sequences, we find a

clear separation between RMs that did or did not develop antibody breadth. Thus, the dynamics of viral fitness may serve as a quantitative signal associated with bnAb development.

The induction of bnAbs is a major goal of HIV-1 vaccine design (*Haynes et al., 2023*). Both computational (*Wang et al., 2015*; *Shaffer et al., 2016*; *Sprenger et al., 2020*; *Nourmohammad et al., 2016*) and experimental (*Dosenovic et al., 2015*; *Escolano et al., 2016*; *Williams et al., 2023*) studies, as well as observations from individuals who developed bnAbs (*Gao et al., 2014*; *Liao et al., 2013*; *Bonsignori et al., 2017*), suggest that the co-evolution of antibodies and HIV-1 is important to stimulate broad antibody responses. Our results could thus inform HIV-1 vaccine research. While precise immune responses and viral escape pathways can differ across individuals, the quantitative similarity in viral evolutionary constraints across humans and RMs suggests that SHIV data can provide a valuable source of information about Env variants that contribute to bnAb development, especially when detailed longitudinal data from humans does not exist. While the concept of sequential immunization is well-established (*Pancera et al., 2010*; *Haynes et al., 2012*; *Klein et al., 2013*; *Wang et al., 2015*; *Escolano et al., 2016*), our findings also suggest a possible new design principle. Immunogens could be engineered to reproduce the dynamics of viral population change that are associated with rapid fitness gains, which we found to precede the emergence of bnAbs. This emphasis on broader, population-level dynamics could complement investigations of the molecular details of virus and antibody coevolution.

As noted above, Roark and collaborators also performed a detailed comparison of HIV-1 and SHIV evolution with the same TF Env sequences (*Roark et al., 2021*). One of their main conclusions was that most Env mutations were selected for escape from CD8$^+$ T cells or antibodies. We found that many antibody resistance mutations identified by Roark et al. are also positively selected in our analysis. Mutations at sites 166 and 169 were shown to confer resistance to a V2 apex bnAb, RHA1, isolated in RM5695 (*Roark et al., 2021*). We inferred moderately positive selection coefficients of 0.49% and 0.43% for R166K and R169K, respectively. The same mutations were found in RM6070, which also developed V2 apex bnAbs, with a selective advantage of 1.7% (*Supplementary file 10*). Mutations conferring resistance to autologous strain-specific nAbs were identified at multiple sites by Roark and colleagues: 130, 234, 279, 281, 302, 330, and 334 in RM6072, which developed antibody responses targeting the CD4 binding site (DH650) and V3 (DH647 and DH648) regions. Mutations Y330H and N334S, which confer resistance to V3 autologous nAbs, were detected in all RMs infected with SHIV. CH505, with selective advantages of 3.0% and 4.6% in RM6072, and 1.7% and 3.2% on average across RMs, respectively. Overall, we found that mutations conferring resistance to autologous strain-specific antibodies were common and more strongly selected than bnAb resistance mutations (*Supplementary file 10* and *Supplementary file 11*).

We note that our conclusions about the phenotypic effects of HIV-1 mutations under selection are constrained by the available data. While we observed strong selection for strain-specific antibody resistance mutations, these results could also be affected by the effects of these mutations on viral replication independent of immune escape. In particular, many ssAb resistance mutations are also reversions to the subtype consensus sequence, which have often been observed to improve viral fitness (*Zanini et al., 2015*; *Sohail et al., 2021*). For example, N334S, K302N, and T234N are all reversions. These are among the most beneficial mutations inferred for SHIV.CH505 (*Supplementary file 8*). In future work, it would be interesting to attempt to fully separate the fitness effects of mutations due to antibody escape and intrinsic replication (*Gao and Barton, 2025*). Although we have systematically compiled information about mutations known to affect antibody resistance and glycosylation, this data is necessarily incomplete. Some of the strongly beneficial mutations with unknown functional effects that we observe could therefore reflect escape from unmapped immune responses.

There are additional methodological and technical limitations that should be considered in the interpretation of our results. Most notably, we assume that the viral fitness landscape is static in time. While we do not expect selection for effective replication ('intrinsic' fitness) to change substantially over time, pressure for immune escape could vary along with the immune responses that drive them. In prior work, we have found that constant selection coefficients typically reflect the average fitness effect of a mutation when its true contribution to fitness is time-varying (*Gao and Barton, 2025*; *Lee et al., 2025*). This may not adequately describe mutational effects that undergo large or rapid shifts in time. Future work should also examine temporal patterns in selection for individual mutations.

While we found a strong relationship between viral fitness dynamics and the emergence of bnAbs, it may not be true that the former stimulates the latter. For example, bnAbs may have been present within each host before they were experimentally detected. Rapid viral fitness gains within hosts that developed broad antibody responses could then have been driven by undetected bnAb lineages. However, we did not find strong selection for known bnAb resistance mutations, and in at least one case (RM5695), rapid fitness gains (roughly 2 weeks after infection) substantially preceded bnAb detection (16 weeks). Still, given the limited size of the data set that we studied, it is unclear the extent to which our results will transfer to larger and broader data sets.

Among other analyses, Roark et al. used LASSIE (*Hraber et al., 2015*) to identify putative sites under selection (*Supplementary file 12* and *Supplementary file 13*). This method works by identifying sites where non-TF alleles reach high frequencies. We found modest overlap between the sites under selection as identified by LASSIE and the mutations that we inferred to be the most strongly selected. For SHIV.CH505, the E640D mutation at site 640 identified by LASSIE is ranked second among 664 mutations in our analysis, and mutations at the remaining 5 sites identified by LASSIE are all within the top 20% of mutations that we infer to be the most beneficial. For SHIV.CH848, the R363Q mutation that is ranked first in our analysis appears at one of the 17 sites identified by LASSIE. Some mutations at the majority of these 17 sites fall within the top 20% most beneficial mutations in our analysis, but some are outliers. In particular, we infer both S291A/P to be somewhat deleterious, with S291P ranked 810th out of 863 mutations.

Beyond the specific context of HIV-1 and bnAb development, our study also provides insight into viral evolution across hosts and related host species. Parallels between the HIV-1 and SHIV fitness landscapes that we infer suggest that there are strong constraints on viral protein function, with few paths to significantly higher fitness. This is consistent with the ideas of methods that use sequence statistics across multiple individuals and hosts to predict the fitness effects of mutations (*Ferguson et al., 2013*; *Mann et al., 2014*; *Lässig et al., 2017*; *Łuksza and Lässig, 2014*; *Barton et al., 2016 Louie et al., 2018 Hie et al., 2021*). However, the relationship between the number of individuals in which a mutation was observed and its inferred fitness effect was fairly weak. This suggests that mutational biases and/or sequence space accessibility may play significant roles in short-term viral evolution, even for highly mutable viruses such as HIV-1 and SHIV. As described above, high-frequency mutations were also not necessarily highly beneficial. While the recombination rate of HIV-1 is high, correlations between mutations persist, making it difficult to unambiguously interpret frequency changes as signs of selection (*Sohail et al., 2021*).

Our results also point to strong similarities in the immune environment across closely related host species, including preferential targeting of specific parts of viral surface proteins by antibodies. This is supported by the enrichment of beneficial mutations within variable loop regions and at sites that affect the glycosylation of Env. However, despite these constraints, there may still exist a large number of neutral or nearly-neutral mutational paths that remain unexplored.

Overall, our findings support the potential predictability of viral evolution, at least over short time scales. While there are contingencies in evolution – for example, disparate host immune responses or strong epistatic constraints between mutations – these are not so pervasive that they completely change the effective viral fitness landscape or paths of evolution across hosts, given the same founder virus sequence. Similar observations of parallel evolution in HIV-1 have been reported in monozygotic twins infected by the same founder virus (*Draenert et al., 2006*), common patterns of immune escape across hosts (*Choisy et al., 2004*; *Barton et al., 2016*) and drug resistance (*Wensing et al., 2016*; *Feder et al., 2014*; *Feder et al., 2016*; *Feder et al., 2021*), and long-term experimental evolution (*Bons et al., 2020*). Our results thus contribute to a growing body of research identifying predictable features in viral evolution. Understanding such features could ultimately inform practical applications such as anticipating the emergence of drug resistance or designing vaccines to limit likely pathways of escape.

## Methods

### Data

We retrieved HIV-1 sequences from CH505 (703010505) and CH848 (703010848) from the HIV sequence database at Los Alamos National Laboratory (LANL) (*Los Alamos National Laboratory,*

*2023a*). The rhesus macaque (RM) SHIV sequences (*Roark et al., 2021*) were obtained from GenBank (*Benson et al., 2012*). We then co-aligned SHIV.CH505 and SHIV.CH848 sequences with CH505 and CH848 HIV-1 sequences, respectively, using HIValign (*Los Alamos National Laboratory, 2023b*).

## CH505

CH505 developed two distinct lineages of CD4 binding site (CD4bs) bnAbs, CH103 and CH235 (*Gao et al., 2014*; *Kreer et al., 2023*). CH103 antibodies were detectable by 14 weeks after infection and further developed neutralization breadth between 41–92 weeks (*Liao et al., 2013*). IC50 values of CH235 against the TF virus were 6.5-fold lower than those of CH103 (*Gao et al., 2014*). CH235 lineages could neutralize autologous viruses at week 30. However, viruses that acquired mutations at loop D from 53 to 100 weeks escaped CH235 neutralization (*Gao et al., 2014*). Although the neutralization breadth of CH235 was not as broad as that of CH103, this lineage played a critical role; escaping mutations from the CH235 lineage stimulated the development of another lineage with broader neutralization depth (*Gao et al., 2014*). Mutations in loop D enabled the virus to escape from CH235, but these sequential mutations in loop D, such as E275K, N279D, and V281S, favorably bound to the mature CH103 and continuously increased the binding affinity between mature CH103 and loop D (*Gao et al., 2014*). Gradually, CH103 matured, developing a broader neutralization breadth.

## CH848

CH848 developed DH270, a bnAb that targets the glycosylated site adjacent to the third variable loop (V3). DH270 was detectable three and a half years after infection (*Bonsignori et al., 2017*). Similar to the CH505 case, the CH848 case exhibited cooperative virus and antibody coevolution. The earlier antibody lineages, DH272 and DH475, could neutralize autologous viruses until week 51 and weeks 15–39, respectively. The virus escaped from DH272 and DH475 afterward, with escape mutations including a longer V1V2 loop. DH270 then developed, with potent and broad neutralization breadth (*Bonsignori et al., 2017*).

## Rhesus macaques

Chimeric viruses, SHIVs, were constructed by bearing the transmitted/founder (TF) Env from three HIV-1 patients, including CH505 and CH848 (*Roark et al., 2021*). In some RMs, SHIV developed similar patterns of mutations to those observed in human donors. In our analysis, we considered RMs with SHIV sequences sampled at at least three points in time. This yielded a set of 7 RMs and 6 RMs for SHIV.CH505 and SHIV.CH848, respectively. The *Supplementary file 6* and *Supplementary file 7* summarize the number of sequences, time points, and the development of bnAbs for each individual in the SHIV cases as well as HIV-1 cases.

## **Sequence data processing**

### Data quality control

To focus our analysis on functional sequences, we removed sequences with more than 200 gaps. To eliminate rare insertions or possible alignment errors, we also masked sites where gaps occurred in more than 95% of sequences within each individual host. To limit errors in virus frequencies, we only considered data from time points with four or more sequences.

### Identifying reversions

A mutation is classified as a reversion if the new (mutant) nucleotide matches with the nucleotide at the same site in the HIV-1 consensus sequence from the same subtype. Here, all viruses were subtype C, so we compared with the subtype C consensus sequence as defined by LANL.

### Identifying mutations that affect N-linked glycosylation

To identify mutations that affect glycosylation, we search for Env mutations that modify the N-linked glycosylation motif Asn-X-Ser/Thr, where X can be any amino acid except proline. We identified three types of mutations affecting glycosylation: 'shields', which complete a previously incomplete glycosylation motif, 'holes', which disrupt an existing glycosylation motif, and 'shifts', which simultaneously complete one N-linked glycosylation motif and disrupt another.

## Enrichment analysis

We used fold enrichment values and Fisher's exact test to quantify the excess or lack of mutations. For a particular subset of mutations (for example, the top $x\%$ beneficial mutations), we first computed the number of mutations in that subset that do ($n_{\text{sel}}$) and do not ($N_{\text{sel}}$) have a particular property (e.g. nonsynonymous mutations in the CD4 binding site). We then computed the total number of mutations that do and do not have the property ($n_{\text{null}}$ and $N_{\text{null}}$, respectively) across the entire data set. The fold enrichment value is then $\frac{n_{\text{sel}}/N_{\text{sel}}}{n_{\text{null}}/N_{\text{null}}}$. The term $\frac{n_{\text{sel}}}{N_{\text{sel}}}$ quantifies the fraction of mutations having specific properties across the selected mutations, while the denominator $\frac{n_{\text{null}}}{N_{\text{null}}}$ is the fraction of all mutations that have the property. Fisher's exact $p$ values are computed from the $2 \times 2$ table with $n_{\text{sel}}$, $n_{\text{null}} - n_{\text{sel}}$ in the first row, and $N_{\text{sel}} - n_{\text{sel}}$, $N_{\text{null}} - N_{\text{sel}} - (n_{\text{null}} - n_{\text{sel}})$ in the second row (**Ruxton and Neuhäuser, 2010**).

## Inferring fitness effects of mutations

In this section, we describe the inference framework used to infer the fitness effects of mutations (selection coefficients) from temporal genetic data.

## Evolutionary model

We model viral evolution with the Wright-Fisher (WF) model, a fundamental model in population genetics (**Ewens, 2004**). In this model, a population of $N$ individuals (viruses or infected cells, in our case) undergoes discrete rounds of selection, mutation, and replication. Each genotype $\alpha$ is represented by a sequence $g^\alpha = ((g^\alpha_{i,a})^q_{a=1})^L_{i=1}$, where $g^\alpha_{i,a}$ is equal to one if genotype $\alpha$ has allele $a$ at locus $i$ and zero otherwise. Here, $L$ and $q$ represent the length of the genetic sequence (number of loci) and the number of statues at each locus (i.e. number of nucleotides or amino acids), respectively. We use $q = 5$ and $q = 21$ for DNA and amino acid sequences, respectively, in real HIV-1 and SHIV data.

We define the fitness of an individual with genetic sequence $g$ by

$$F(g) = 1 + \sum_{i=1}^{L}\sum_{a=1}^{q} s_i(a)g_{i,a}. \tag{4}$$

Here $s_i(a)$ is a selection coefficient, quantifying the fitness effect of allele $a$ at locus $i$. If $s_i(a) > 0$, the allele $a$ is beneficial (enhancing replication), and if $s_i(a) < 0$ it is deleterious (impairing replication). By convention, we set the selection coefficient for TF alleles to zero. Individuals with higher fitness values are more likely to replicate than those with lower fitness.

Mutations introduce new genotypes and drive the evolution of the population. Let us define $\mu^{\alpha\beta}$ as the probability of mutation from genotype $\alpha$ to genotype $\beta$ per replication cycle. Below, we will express this probability in terms of a mutation rate per site per round of replication. In the analysis of real data, we use asymmetric mutation rates estimated from intra-host HIV-1 data (**Zanini et al., 2015**).

Given these parameters, the WF model describes the dynamics of the frequencies of different genotypes in the population over time. We write the frequency of genotype $\alpha$ at time $t$ as $z_\alpha(t)$. Given that the frequency of genotypes in the population at time $t$ is $z(t) = (z_1(t), z_2(t), \ldots, z_M(t))$, where $M$ is the total number of genotypes, the probability distribution of the frequency of genotypes in the next generation $z(t + 1)$ is

$$P(z(t+1)|z(t); s, \mu, N) = \prod_{a=1}^{M}\left(\frac{p_\alpha(z(t))^{Nz_\alpha(t+1)}}{[Nz_\alpha(t+1)]!}\right). \tag{5}$$

$p_\alpha$ here is

$$p_\alpha(z(t)) \propto F^\alpha z_\alpha + \sum_{\beta(\neq\alpha)}[\mu^{\beta\alpha}z_\beta(t) - \mu^{\alpha\beta}z_\alpha(t)]. \tag{6}$$

where $F^\alpha$ is the fitness value of genotype α, based on **Equation 4**. Across $K$ generations, the probability of an entire evolutionary trajectory, defined by the vector of genotype frequencies at each time, is then

$$P((z(t_k))_{k=0}^{K}|s, \mu, N) = \prod_{k=0}^{K-1} P(z(t_{k+1})|z(t_k); s, \mu, N). \tag{7}$$

## Diffusion limit

When the population size is sufficiently large, the evolution of the population defined in **Equation 5** can be reasonably well approximated by a Gaussian process, which is a solution to the Fokker-Planck (forward Kolmogorov) equation (**Kimura, 1964**; **Crow, 2017**).

$$P(z(t + \Delta t)|z(t); s, \mu, N) \sim \mathcal{N}\left(z(t) + \Delta t d(t), C(z(t))/N\right), \tag{8}$$

with the drift vector $d(t)$ and the diffusion matrix $C(z(t))/N$ such that

$$C_{\alpha\beta}(z(t)) = \begin{cases} -z_\alpha(t)z_\beta(t) & \text{for } \alpha \neq \beta \\ z_\alpha(t)(1 - z_\alpha(t)) & \text{for } \alpha = \beta, \end{cases} \tag{9}$$

and

$$d(t) = C(z(t))s + u. \tag{10}$$

## Dimensional reduction

While the WF process in genotype space provides valuable insights into genotype dynamics, the mathematical expressions are sometimes challenging to interpret. To obtain more intuitive expressions, we can project the dynamics onto the space of allele frequencies,

$$x_i(a) = \sum_a g_{i,a}^\alpha z_\alpha. \tag{11}$$

One can then find the drift vector $d$ and diffusion matrix $C/N$ in allele frequency space,

$$d_i(a) = C_{ii}(a, a)s_i(a) + \sum_{j(\neq i)}^{L} \sum_{b=1}^{q} C_{ij}(a, b)s_j(b) + u_i(a), \tag{12}$$

and

$$C_{ij}(a, b) = \begin{cases} x_{ij}(a, b) - x_i(a)x_j(b) & \text{for } i \neq j \\ x_i(a)(1 - x_i(a)) & \text{for } i = j. \end{cases} \tag{13}$$

Here, $x_{ij}(a, b)$ is the frequency of individuals with alleles $a$ and $b$ at loci $i$ and $j$, and $u_i(a)$ is net expected change in frequency of allele $a$ at $i$ due to mutations, which is given explicitly in **Equation 17** below. The first term in **Equation 12** gives the expected change in the frequency $x_i(a)$ due to the direct fitness effect $s_i(a)$, while the second term represents the contributions due to indirect or genetic linkage effects with other alleles $j$.

## Maximum path likelihood

Following recent work (**Sohail et al., 2021**), we employed Bayes' rule to find the selection coefficients that best explain the data. These are the coefficients that maximize the posterior distribution

$$P_{\text{posterior}}(s|(z(t_k))_k)((z(t_k))_k|s) P_{\text{prior}}(s), \tag{14}$$

which is a product of the likelihood of the evolutionary trajectory observed in the data **Equation 7** (under the diffusion limit **Equation 8**) and a prior distribution for the selection coefficients. We chose a Gaussian prior distribution with zero mean and a covariance of $I/(N\gamma)$, where $I$ is the identity matrix. This prior distribution penalizes the inferences of large selection coefficients when they are not well-supported by the data. The maximum a posteriori selection coefficients are then given by **Sohail et al., 2021**

$$\hat{s} = \left(C_{\text{int}} + \gamma I\right)^{-1} \left[\Delta x_{\text{int}} - \Delta u_{\text{int}}\right].$$ (15)

Here $C_{\text{int}}$, $_{\text{int}}$, and $_{\text{int}}$ represent the covariance matrix, vector of frequency changes, and mutational flux integrated over the evolution

$$\begin{aligned}
C_{\text{int}} &= \sum_{k=0}^{K} \Delta t_k C(x(t_k)) \\
\Delta x_{\text{int}} &= x(t_K) - x(t_0) \\
\Delta u_{\text{int}} &= \sum_{k=0}^{K-1} \Delta t_k u(t_k).
\end{aligned}$$ (16)

The mutational flux $u$ is characterized by the rates of mutations from nucleotides $b$ to $a$, denoted by $\mu_{ab}$, which are determined from longitudinal HIV-1 populations in untreated patients (*Zanini et al., 2015*). The change of the $a$ nucleotide frequency at locus $i$ due to mutation is given by:

$$u_i(a) = \sum_{b} \left(\mu_{ab} x_i(b) - \mu_{ba} x_i(a)\right).$$ (17)

Inverting the integrated covariance matrix effectively reveals the underlying direct allele interactions and resolves the genetic linkage effects.

The shift in the covariance diagonal in *Equation 15*, arising from the selection coefficients' posterior distribution, reflects the uncertainty in the selection distribution. We used $\gamma = 10$ for all data sets, but the model is robust to variation in the strength of regularization (*Sohail et al., 2021*). For the mutation rates, we incorporated the transition probabilities among arbitrary DNA nucleotides, estimated from whole-genome deep sequencing of multiple untreated HIV-1 patients followed for 5–8 years post-infection (*Zanini et al., 2015*).

## Integration of covariance

When the time interval of the observation $\Delta t$ is sufficiently short, the trajectory of the allele frequency would be continuous and ideally it would be a smooth curve (*Shimagaki and Barton, 2023*). To accurately estimate the covariance matrix, we employ piecewise linear interpolation for frequencies. Let $\tau \in [0, 1]$, then the linear interpolation for a frequency vector can be expressed as:

$$\begin{aligned}
x_i^{[k,k+1]}(\tau) &= (1 - \tau)x_i(t_k) + \tau x_i(t_{k+1}), \\
x_{ij}^{[k,k+1]}(\tau) &= (1 - \tau)x_{ij}(t_k) + \tau x_{ij}(t_{k+1}),
\end{aligned}$$ (18)

which yields,

$$x_i^{\text{int}} = \sum_{k=0}^{K-1} \Delta t_k \int_0^1 d\tau \, x_i^{[k,k+1]}(\tau)$$

$$C_{ij}^{\text{int}} = \sum_{k=0}^{K-1} \Delta t_k \int_0^1 d\tau \, \left(x_{ij}^{[k,k+1]}(\tau) - x_i^{[k,k+1]}(\tau) x_j^{[k,k+1]}(\tau)\right)$$

For simplicity in notation, we omitted nucleotide indices. The explicit expression of the integrated covariance is given in *Sohail et al., 2021*; *Shimagaki and Barton, 2023*.

## Joint RM model

In addition to fitness models derived from SHIV data for individual RMs, we inferred a joint model under the assumption that virus evolution within each individual RM with the same TF virus is governed by a similar fitness landscape. This method improves inference accuracy from the WF process and deep mutational scanning data (*Sohail et al., 2022*; *Shimagaki and Barton, 2025b*; *Hong et al., 2024*). The joint path likelihood for allele frequency trajectories across RMs with the same TF virus is then

$$p(((x^\alpha(t_k))_{k=0}^{K_\alpha})_{r=1}^R | s, \mu, \gamma) = \left(\prod_{r=1}^{R} \prod_{k=0}^{K-1} p_{\text{M}}(x^r(t_{k+1}) | x^r(t_k); s, \mu, N)\right) p(x(t_0)) \, p(s|\gamma).$$ (19)

Here, $x^r$ is the allele frequency of the $r$-th individual and $R$ is the number of replicate individuals (i.e. the number of RMs sharing the same TF virus). The initial state is $p(x^r(t_0)) = p(x(t_0))$ for all $r$ for individuals with the same TF virus. The solution of the joint path likelihood is given by

$$\hat{s} = (\overline{C}_{\text{int}} + \gamma I)^{-1}[\overline{\Delta x}_{\text{int}} - \overline{\Delta a \mu}_{\text{int}}].$$ 

(20)

Here, the overbar denotes the sum over the replicate RMs.

We emphasize that the joint selection coefficients in *Equation 20* are not the same as selection coefficients that are simply averaged across RMs with the same TF virus. The joint selection coefficients are more robust, as they are guided by the level of evidence within each individual rather than naive averaging.

## Geometrical interpretation of the fitness comparison

The Pearson values we utilized to compare the fitness landscapes denoted as $F_s(g) := F(g|s)$ and $F_h(g) := F(g|h)$ can be expressed by the following simple relation:

$$r = \frac{\text{Cov}(F_s, F_h)}{\sqrt{\text{Var}(F_s)}\sqrt{\text{Var}(F_h)}} = \frac{s^\top C h}{\sqrt{s^\top C s}\sqrt{h^\top C h}}.$$ 

(21)

Here, $\text{Cov}(F_s, F_h)$ and $\text{Var}(F_s)$ represent the covariance and variance values estimated from the samples being compared, $(F_s(g^n), F_h(g^n))_n$. $C$ is the covariance matrix defined between arbitrary loci. The last equation can be interpreted as an angle between two vectors, $s$ and $h$, with a metric matrix $C$; if $s = h$, the Pearson value clearly becomes 1. However, the 'similarity' also depends on how these vectors are projected by $C$; eigenmodes associated with larger variance of statistics will be more emphasized. The last expression readily implies an interpretation for the case of shuffled sequences; shuffling the sequences equates to diluting the covariance between loci, resulting in the metric matrix becoming a diagonal matrix. Removing the off-diagonal elements corresponds to lifting the constraints on the fitness landscape.

## Epistatic fitness model

We consider the following pairwise epistatic fitness function, which depends on epistatic interactions $s_{ij}$ across all possible pairs of loci:

$$F(g) = 1 + \sum_{i=1}^{L} \sum_{a=1}^{q} s_i(a) g_{i,a} + \sum_{i<j} \sum_{a,b=1}^{q} s_{ij}(a,b) g_{i,a} g_{j,b}.$$ 

(22)

Our goal is to obtain the epistatic interactions $s_{ij}(a,b)$ as well as the selection coefficients $s_i(a)$ from temporal genetic sequences.

The basic logic for inferring these fitness parameters parallels the additive case. The only practical difference is that epistatic interactions can influence the dynamics of additive and pairwise mutation frequencies. Under the diffusion limit, we obtain the following drift terms, which align with (12) in the additive model,

$$
\begin{aligned}
d_i(a) &= \sum_{k=1}^{L} \sum_{c=1}^{q} C_{ik}(a,c) s_k(c) + \sum_{k<l} \sum_{c,d=1}^{q} C_{ikl}(a,c,d) s_{kl}(c,d) + u_i(a) \\
d_{ij}(a,b) &= \sum_{k}^{L} \sum_{c=1}^{L} C_{ijk}(a,b,c) s_k(c) + \sum_{k<l}^{L} \sum_{c,d=1}^{q} C_{ijkl}(a,b,c,d) s_{kl}(c,d) + u_{ij}(a,b) + v_{ij}(a,b).
\end{aligned}
$$ 

(23)

and diffusion matrices,

$$C_{ikl}(a,c,d) = x_{ikl}(a,c,d) - x_i(a) x_{kl}(c,d)$$

$$C_{jikl}(a,b,c,d) = x_{ijkl}(a,b,c,d) - x_{ij}(a,b) x_{kl}(c,d).$$ 

(24)

Here, $u_i$ and $u_{ij}$ represent the expected frequency changes due to mutations for additive and pairwise terms, while $v_i$ represents the changes in pairwise frequencies due to recombination. These explicit expressions indicate that the $u_i$ remains the same as in the additive fitness case; therefore, *Equation 17* holds. The pairwise term is given as

$$u_{ij}(a,b) \quad = \sum_{c=1}^{q} \Big( [\mu_{bc} x_{ij}(a,c) + \mu_{ac} x_{ij}(c,d)] - [\mu_{cd} + \mu_{ca}] x_{ij}(a,b) \Big). \tag{25}$$

and the $v_{ij}$ is expressed as

$$v_{ij}(a,b) = -r|i - j| C_{ij}(a,b), \tag{26}$$

where $r$ denotes the recombination rate. In this study, we set $r = 10^{-5}$. More detailed derivations are provided in previous studies (*Sohail et al., 2022*; *Shimagaki and Barton, 2025b*).

The technical challenge of the epistasis inference is that the diffusion matrix $C$ involves third- and fourth-order interactions, and the number of matrix elements scales as $\mathcal{O}((qL)^4)$, while the computational cost to invert it scales as $\mathcal{O}((qL)^6)$. Recently, a more efficient computational method was proposed, reducing both the necessary memory usage and computational times by $\mathcal{O}((qL)^2)$ (*Shimagaki and Barton, 2025b*). The essential idea involves factorizing the higher-order covariance matrix using the rectangular matrix $\Xi \in \mathbb{R}^D$ such that $C = \Xi \Xi^\top$, where $D$ scales as $\mathcal{O}((qL)^2)$ while $d$. This method resolves the linear equation without obtaining an explicit expression of $C$ and avoids any computations involving more than $\mathcal{O}((qL)^2)$ operations (*Shimagaki and Barton, 2025b*).

## Gauge transformation

Since constant shifts in fitness parameters do not affect relative fitness, it is always possible to transform fitness values without changing the resulting genotype distribution. For example, in the additive model, individual selection coefficients can be shifted as

$$s_i(a) \leftarrow s_i(a) - s_i(\tilde{g}_i),$$

where $\tilde{g}_i$ is the allele of a chosen reference genotype $\tilde{g}$ (e.g. the TF sequence) at site $i$. This transformation preserves relative fitness, but it ensures that $s_i(\tilde{g}_i) = 0$, making inferred selection coefficients more interpretable.

In statistical physics, such transformations, where model parameters are changed without altering the underlying probability distribution, are referred to as gauge transformations (*Weigt et al., 2009*; *Morcos et al., 2011*). Similar transformations have been employed in recent studies to improve interpretability and sparsity in epistatic models (*Rizzato et al., 2020*). We can apply an analogous transformation to epistatic interactions:

$$\begin{aligned} s_i(a) &\leftarrow s_i(a) - s_i(\tilde{g}_i) + \sum_{j|j\neq i} \left[ s_{ij}(a,\tilde{g}_j) - s_{ij}(\tilde{g}_i,\tilde{g}_j) \right], \\ s_{ij}(a,b) &\leftarrow s_{ij}(a,b) - s_{ij}(\tilde{g}_i,b) - s_{ij}(a,\tilde{g}_j) + s_{ij}(\tilde{g}_i,\tilde{g}_j). \end{aligned} \tag{27}$$

Under this transformation, any selection coefficients or epistatic terms involving TF alleles are zero by definition, while relative fitness remains unchanged.

## Regularization

Regularization is used to reduce the effective number of parameters in the fitness model. In our analysis, we applied strong regularization ($\gamma = 10^{10}$) to any selection or epistatic coefficients involving TF alleles, ensuring they are effectively zero under the gauge transformation. Following prior work (*Shimagaki and Barton, 2025b*), we penalized epistatic interactions between loci more than 50 nucleotides apart on the reference sequence with the same strong regularization. We also used the same moderate regularization value of $\gamma = 50$ for all other epistatic terms and used $\gamma = 10$ for selection coefficients, consistent with the additive model.

## Acknowledgements

The work of KSS and JPB reported in this publication was supported by the National Institute of General Medical Sciences of the National Institutes of Health under Award Number R35GM138233.

# Additional information

## Funding

| Funder | Grant reference number | Author |
|---|---|---|
| National Institutes of Health | R35GM138233 | Kai S Shimagaki John P Barton |

The funders had no role in study design, data collection and interpretation, or the decision to submit the work for publication.

## Author contributions

Kai S Shimagaki, Conceptualization, Data curation, Investigation, Visualization, Methodology, Writing – original draft, Writing – review and editing; Rebecca M Lynch, Investigation, Writing – original draft, Writing – review and editing; John P Barton, Conceptualization, Data curation, Funding acquisition, Investigation, Visualization, Methodology, Writing – original draft, Project administration, Writing – review and editing

## Author ORCIDs

Kai S Shimagaki ⬤ https://orcid.org/0000-0001-7580-4781
Rebecca M Lynch ⬤ https://orcid.org/0000-0002-7188-8073
John P Barton ⬤ https://orcid.org/0000-0003-1467-421X

Reviewer #1 (Public review): https://doi.org/10.7554/eLife.105466.3.sa1
Reviewer #2 (Public review): https://doi.org/10.7554/eLife.105466.3.sa2
Reviewer #3 (Public review): https://doi.org/10.7554/eLife.105466.3.sa3
Author response https://doi.org/10.7554/eLife.105466.3.sa4

# Additional files

## Supplementary files

Supplementary file 1. CH103 and CH235 resistance mutations.

Supplementary file 2. Strain-specific antibody resistance mutations in CH505.

Supplementary file 3. DH272, DH475, and strain-specific antibody resistance mutations in CH848.

Supplementary file 4. Biological effects of the HIV-1 mutations inferred to be the most beneficial in CH505.

Supplementary file 5. Biological effects of the HIV-1 mutations inferred to be the most beneficial in CH848.

Supplementary file 6. CH505 and SHIV.CH505 sequence statistics.

Supplementary file 7. CH848 and SHIV.CH848 sequence statistics.

Supplementary file 8. Biological effects of strongly selected SHIV.CH505 mutations.

Supplementary file 9. Biological effects of strongly selected SHIV.CH848 mutations.

Supplementary file 10. Selective advantage of mutations that confer resistance to antibodies in SHIV. CH505.

Supplementary file 11. Selective advantage of mutations that confer resistance to antibodies in SHIV. CH848.

Supplementary file 12. List of selected sites using LASSIE in SHIV.CH505.

Supplementary file 13. List of selected sites using LASSIE in SHIV.CH848.

MDAR checklist

## Data availability

Data and code accompanying this manuscript is publicly available at the GitHub repository https://github.com/bartonlab/paper-HIV-coevolution (copy archived at *Shimagaki and Barton, 2025a*). This repository contains source files that process HIV-1 and SHIV sequences, infer selection coefficients,

and identify and characterize mutations. The included Jupyter notebooks can be run to reproduce the figures presented here. The original HIV-1 sequences can be retrieved from the LANL database (https://www.hiv.lanl.gov/content/index), and SHIV sequences can be found at GenBank (https://www.ncbi.nlm.nih.gov/).

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
