## [Editor Report · eLife Assessment]

In this **important** quantitative study of HIV-1 evolution in humans and rhesus macaques, selection coefficients are inferred at scale over the HIV genome. Selection coefficients are similar in humans and macaques, providing **compelling** evidence that these coefficients are representative of the fitness landscapes of these viruses within hosts. This work will be of interest to the community working on quantitative evolution and fitness landscape inference, and the finding that rapid fitness gains in the HIV population predict bNAb emergence has significant implications for HIV vaccine design.

---

## [Referee Report · Reviewer #1 (Public review)]

Summary:

The present work studies the coevolution of HIV-1 and the immune response in clinical patient data. Using the Marginal Path Likelihood (MPL) framework, they infer selection coefficients for HIV mutations from time-series data of virus sequences as they evolve in a given patient.

Strengths:

The authors analyze data from two human patients, consisting of HIV population sequence samples at various points in time during the infection. They inferred selection coefficients from the observed changes in sequence abundance using MPL. Most beneficial mutations appear in viral envelop proteins. The authors also analyze SHIV samples in rhesus macaques, and find selection coefficients that are compatible with those found in the corresponding human samples.

The manuscript is well written and organized.

Comments on revisions:

In their revised version the authors have addressed most of these points satisfactorily.

---

## [Referee Report · Reviewer #2 (Public review)]

This paper combines a biological topic of interest with the demonstration of important theoretical/methodological advances. Fitness inference is the foundation of the quantitative analysis of adapting systems. It is a hard and important problem and this paper highlights a compelling approach (MPL) first presented in (1) and refined in (2), roughly summarized in equation 3.

The authors find that positive selection shapes the variable regions of env in shared patterns across two patient donors. The patterns of positive selection are interesting in and of themselves, they confirm the intuition that hyper-variation in env is the result of immune evasion rather than a broadly neutral landscape (flatness). They show that the immune evasion patterns due to CD8 T and naive B-cell selection are shared across patients. Furthermore, they suggest that a particular evolutionary history (larger flux to high fitness states) is associated with bNAb emergence. Mimicking this evolutionary pattern in vaccine design may help us elicit bNAbs in patients in the future.

The fitness landscape of env in multiple hosts is immensely valuable especially because of how often SHIV has used as proxy for HIV. The strength of reversion-to-consensus selection is a known pattern of HIV post-infection populations but they are nicely quantified here. Agreement between SHIV and HIV evolution is shown. They find selection is larger for autologous antibodies than the bNAbs themselves (perhaps bNAbs are just too small a component of the host response to drive the bulk of selection?), and that big fitness increases precede antibody breadth in rhesus-macaques, suggesting that this fitness increase is the immune challenge required to draw forth a bNAb. All of high interest to HIV researchers.

(1) Sohail, M. S., Louie, R. H., McKay, M. R. & Barton, J. P. Mpl resolves genetic linkage in fitness inference from complex evolutionary histories. Nature biotechnology 39, 472-479 (2021).

(2) Shimagaki, K. & Barton, J. P. Bézier interpolation improves the inference of dynamical models from data. Physical Review E 107, 024116 (2023).

Strength of evidence:

Equation 3 is a beautiful and intuitive tool that accounts for linkage and can be solved precisely even in the presence of detailed mutational and selection models. They have addressed my earlier concerns the effects of incomplete observations of the frequency bias fitness inference on rare sites.

Whether the fact that fitness increases occured before or after the presence of the bnab remains incompletely known. bNAb detection is different from bNAb presence and the possibility that fitness increases occurred after the bNAbs appeared remains. Still, their conclusion is plausible and fits in with the other observations which form a coherent and compelling picture.

Overall this is a convincing paper. It is a valuable introduction to a practical method of fitness inference at the scale of the entire env gene and how this information can be leveraged to learn some interesting biology.

---

## [Referee Report · Reviewer #3 (Public review)]

Summary:

Shimagaki et al. investigate the virus-antibody coevolutionary processes that drive the development of broadly neutralizing antibodies (bnAbs). The study's primary goal is to characterize the evolutionary dynamics of HIV-1 within hosts that accompany the emergence of bnAbs, with a particular focus on inferring the landscape of selective pressures shaping viral evolution. To assess the generality of these evolutionary patterns, the study extends its analysis to rhesus macaques (RMs) infected with simian-human immunodeficiency viruses (SHIV) incorporating HIV-1 Env proteins derived from two human individuals.

Strengths:

A key strength of the study is its rigorous assessment of the similarity in evolutionary trajectories between humans and macaques. This cross-species comparison is particularly compelling, as it quantitatively establishes a shared pattern of viral evolution using a sophisticated inference method. The finding that similar selective pressures operate in both species adds robustness to the study's conclusions and suggests broader biological relevance. In the revised version, the Authors included a simple but clear explanation of the statistical method for inferring the model's parameters in the main text. Moreover, I find the potential implications of the methodology absent in the original submission very interesting.

Conclusions:

Overall, the study presents a compelling analysis of HIV-1 evolution and its parallels in SHIV-infected macaques.

---

## [Author Response]

The following is the authors’ response to the original reviews.

**Reviewer #1 (Public review):**
Summary:The present work studies the coevolution of HIV-1 and the immune response in clinical patient data. Using the Marginal Path Likelihood (MPL) framework, they infer selection coefficients for HIV mutations from time-series data of virus sequences as they evolve in a given patient.Strengths:The authors analyze data from two human patients, consisting of HIV population sequence samples at various points in time during the infection. They infer selection coefficients from the observed changes in sequence abundance using MPL. Most beneficial mutations appear in viral envelop proteins. The authors also analyze SHIV samples in rhesus macaques, and find selection coefficients that are compatible with those found in the corresponding human samples.Weaknesses:The MPL method used by the authors considers only additive effects of mutations, thus ignoring epistasis.

As suggested, we have now addressed this limitation by inferring epistatic fitness landscapes for CH505, CH848, SHIV.CH505, and SHIV.CH848. Indeed, the computational burden of the epistasis inference procedure was one constraint that motivated us to consider only additive fitness in the previous version of our paper. The original approach developed by Sohail et al. (2022) tested only sequences with <50 sites due to this limitation, far smaller than the ones we consider. Beyond this computational constraint, we also believed that (1) an additive fitness model may suffice to capture local fitness landscapes, and practically, (2) epistatic interactions are more challenging to validate than the effects of individual mutations, making the interpretation of the model more complex.

However, after performing the analyses described in this paper, we developed a new approach for identifying epistatic interactions that can scale to much longer sequences (Shimagaki et al., Genetics, in press). We therefore applied this method to infer an epistatic fitness landscape for the HIV and SHIV data sets that we studied. As in that work, we focused on short-range (<50 bp) interactions which we could more confidently estimate from data. We have added a section in the SI describing the epistatic fitness model and our analysis.

Overall, we found substantial agreement between the epistatic and purely additive models in terms of the estimated fitness effects of individual mutations (new Supplementary Fig. 8) and overall fitness (Supplementary Fig. 9). Consistent with our prior work, we did not find substantial evidence for very strong epistatic interactions (Supplementary Fig. 10). This does not necessarily mean that strong epistatic interactions do not exist; rather, this shows that strong interactions don’t substantially improve the fit of the model to data, and thus many are regularized toward zero. While the biological validation of epistatic interactions is challenging, we found that the largest epistatic interactions, which we defined as the top 1% of all shortrange interactions, were modestly but significantly enriched in the CD4 binding site, V1 and V5 regions for CH505 and in the CD4 binding site, V4, and V5 for CH848. In addition, mutation pairs N280S/V281A and E275K/V281G, which confer resistance to CH235, ranked in the top 15% of all epistatic interactions in CH505.

We have now included an additional section in the Results, “Robustness of inferred selection to changes in the fitness model and finite sampling”, which discusses our epistatic analyses (page 6, lines 415-464), along with the above Supplementary Figures and a technical section in the SI summarizing the epistasis inference approach.

Although the evolution of broadly neutralizing antibodies (bnAbs) is a motivating question in the introduction and discussion sections (and the title), the relevance of the analysis and results to better understanding how bnAbs arise is not clear. The only result presented in direct connection to bnAbs is Figure 6.

It is true that, while bnAb development is a major motivator of our study, our analysis focuses on HIV-1 and does not directly consider antibody evolution. We have now brought attention to this point as a limitation directly in the Discussion. Following the suggestion below in the “Recommendations for the authors,” we have edited our manuscript to place more emphasis on viral fitness and somewhat reduce the emphasis on bnAbs, though this remains an important motivating factor. Specifically, the Abstract now begins

Human immunodeficiency virus (HIV)-1 evolves within individual hosts to escape adaptive immune responses while maintaining its capacity for replication. Coevolution between the HIV-1 and the immune system generates extraordinary viral genetic diversity. In some individuals, this process also results in the development of broadly neutralizing antibodies (bnAbs) that can neutralize many viral variants, a key focus of HIV-1 vaccine design. However, a general understanding of the forces that shape virusimmune coevolution within and across hosts remains incomplete. Here we performed a quantitative study of HIV-1 evolution in humans and rhesus macaques, including individuals who developed bnAbs.

We have similarly modified the Discussion to focus first on viral fitness. In response to comments from Reviewer 3, we have also more clearly articulated how our work might contribute to the understanding of bnAb development in the Discussion.

Questions or suggestions for further discussion:I list here a number of points for which I believe the paper would benefit if additional discussion/results were included.The MPL method used by the authors considers only additive effects of mutations, thus ignoring epistasis. In Sohail et al (2022) MBE 39(10), p. msac199 (https://doi.org/10.1093/molbev/msac199) an extension of MPL is developed allowing one to infer epistasis. Can the authors comment on why this was not attempted here?I presume one possible reason is that epistasis inference requires considerably more computational effort (and more data). However, since the authors find most beneficial mutations occurring in Env, perhaps restricting the analysis to Env genes only (e.g. the trimer shown in Figure 2) can lead to tractable inference of epistasis within this segment (instead of the full genome).

As described above, we have now addressed this comment by inferring epistatic fitness landscapes for the data sets that we consider. Our overall results using the epistatic fitness model are consistent with the ones that we previously obtained with an additive model.

Do the authors find correlations in the inferred selection coefficients of the two samples CH505 and CH848? I could not find any discussion of this in the manuscript. Only correlations between Humans and RM are discussed.

To address this question, we compared the fitness values and individual selection coefficients across CH505 and CH848 data sets. We found little correlation between CH505 and CH848 fitness values (shown in a new Supplementary Fig. 6) or selection coefficients. We found only 199 common mutations between HIV-1 amino acid sequences from CH505 and CH848 out of 868 and 1,406 total mutations, respectively. Thus, we were not surprised to find no strong relationship between fitness estimates from CH505 and CH848 data sets.

**Reviewer #2 (Public review):**
Summary:This paper combines a biological topic of interest with the demonstration of important theoretical/methodological advances. Fitness inference is the foundation of the quantitative analysis of adapting systems. It is a hard and important problem and this paper highlights a compelling approach (MPL) first presented in (1) and refined in (2), roughly summarized in equation 12.(1) Sohail, M. S., Louie, R. H., McKay, M. R. & Barton, J. P. Mpl resolves genetic linkage in fitness inference from complex evolutionary histories. Nature biotechnology 39, 472-479 (2021).(2) Shimagaki, K. & Barton, J. P. Bézier interpolation improves the inference of dynamical models from data. Physical Review E 107, 024116 (2023).The authors find that positive selection shapes the variable regions of env in shared patterns across two patient donors. The patterns of positive selection are interesting in and of themselves, they confirm the intuition that hyper-variation in env is the result of immune evasion rather than a broadly neutral landscape (flatness). They show that the immune evasion patterns due to CD8 T and naive B-cell selection are shared across patients. Furthermore, they suggest that a particular evolutionary history (larger flux to high fitness states) is associated with bNAb emergence. Mimicking this evolutionary pattern in vaccine design may help us elicit bNAbs in patients in the future.There is a lot of information to be found in the full fitness landscape of env. The enormous strength of reversion-to-consensus in the patterns is a known pattern of HIV post-infection populations but they are nicely quantified here. Agreement between SHIV and HIV evolution is shown. They find selection is larger for autologous antibodies than the bNAbs themselves (perhaps bNAbs are just too small a component of the host response to drive the bulk of selection?), and that big fitness increases precede antibody breadth in rhesus macaques, suggesting that this fitness increase is the immune challenge required to draw forth a bNAb. This is all of high interest to HIV researchers.Strength of evidence:One limitation is, of course, that the fitness model is constant in time when the immune challenge is variable and changing. This simplification may complicate some interpretations.

We agree that this is a limitation of our current approach. In prior work, we have found that the constant fitness effects of mutations that we infer typically reflect the time-averaged fitness effect when the selection changes over time (Gao and Barton, PNAS 2025; Lee et al., Nat Commun 2025). It could be difficult, however, to capture changes in selection that fluctuate rapidly with underlying immune responses. We have added a new paragraph in the Discussion that more clearly sets out some of the limitations of our analysis, including our assumption of constant selection coefficients.

There are additional methodological and technical limitations that should be considered in the interpretation of our results. Most notably, we assume that the viral fitness landscape is static in time. While we do not expect selection for effective replication (“intrinsic” fitness) to change substantially over time, pressure for immune escape could vary along with the immune responses that drive them. In prior work, we have found that constant selection coefficients typically reflect the average fitness effect of a mutation when its true contribution to fitness is time-varying [42,43]. This may not adequately description mutational effects that undergo large or rapid shifts in time. Future work should also examine temporal patterns in selection for individual mutations.

Equation 12 in the methods is really a beautiful tool because it is so simple, but accounts for linkage and can be solved precisely even in the presence of detailed mutational and selection models. However, the reliance on incomplete observations of the frequency leads to complications that must be carefully (re)addressed here.For instance, the consistent finding of strong selection in hypervariable regions is biologically intuitive but so striking, that I worry that it might be the result of a bias for selection in high entropy regions.

Thank you for this suggestion. We agree that it is important to carefully interrogate these results. To assess the effects of general sequence variability on inferred selection, we first computed a position-specific entropy measure, H_i_, for each site i. We first defined the time-dependent entropy H_i_(t) = - ∑_a_ x_i_ (a, t) log x_i_ (a, t), where x_i_ (a, t) represents the frequency of amino acid/nucleotide a at position i and time t, at each sample time. We then computed H_i_ as the average of H_i_(t) across all sample times. A new Supplementary Fig. 1 plots the entropy against the inferred selection coefficients. Although some sequence variation must be observed in order for us to infer that a mutation is beneficial, we did not find a systematic bias toward larger (more beneficial) selection coefficients at more variable sites. Overall, we found only a modest correlation between inferred selection coefficients and entropy (Pearson’s r = 0.33 and 0.29 for CH505 and CH848, respectively), which appears to be partly driven by the tendency for mutations inferred to be significantly deleterious to occur at sites with low entropy. In addition to the new Supplementary Figure, we have added a reference to this analysis in the main text:

To test whether our results might be biased by overall sequence variability, we examined the relationship between our inferred selection coefficients and entropy, a common measure of sequence variability. Overall, we found only a modest correlation between selection and entropy, suggesting that the signs of selection that we observe are not due to increased sequence variability alone (Supplementary Fig. 1).

Mutational and covariance terms in equation 12 might be underestimated, due to finite sampling effect in highly diverse populations. Sampling effects lead to zeros in x(t) when actual frequency zeros might be rare at the population sizes of HIV viral loads and mutation rates. Both mutational flux and C underestimation will bias selection upward in eq. 12.The prior papers (1) and (2) seem to show robustness to finite sampling effects, but, again, more care needs to be shown that this robustness transfers to the amino acid inference under these conditions. That synonymous sites are rarely selected for in the nucleotide level is a good sign, and it may be a matter of simply fully explaining the amino-acid level model.

As above, we agree that these tests are important. To assess the robustness of our results to finite sampling, we performed bootstrap sampling on the viral sequences and inferred selection coefficients using the resampled sequences. Specifically, we resampled the same number of sequences as in the original data at each time point and repeated this for all time points across all HIV-1 and SHIV data sets. A new Supplementary Fig. 11 shows a typical comparison of the original selection coefficients vs. those obtained through bootstrap resampling. Overall, we observe a high degree of consistency between the selection coefficients in each case, which is surely aided by the long time series in these data sets. As pointed out by the reviewer, uncertainty in low-frequency mutations is a particular concern, though the effects on inferred selection are mitigated by regularization.

We have added a section in the Results, “Robustness of inferred selection to changes in the fitness model and finite sampling”, which includes this analysis:

Finite sampling of sequence data could also affect our analyses. To further test the robustness of our results, we inferred selection coefficients using bootstrap resampling, where we resample sequences from the original ensemble, maintaining the same number of sequences for each time point and subject. The selection coefficients from the bootstrap samples are consistent with the original data (see Supplementary Fig. 11), with Pearson’s r values of around 0.85 for HIV-1 data sets and 0.95 for SHIV data sets, respectively.

Uncertainty propagates to the later parts of the paper, eg. HIV and SIV shared patterns might be the result of shared biases in the method application. However, this worry does not extend to the apples-to-apples comparison of fitness trajectories across individuals (Figures 5 and 6) which I think are robust (for these sample sizes).

One way to address this uncertainty is to compare the fitness values and individual selection coefficients across CH505 and CH848 data sets, which was also requested by Reviewer 1. Overall, we found little correlation between CH505 and CH848 fitness values (shown in a new Supplementary Fig. 6) or selection coefficients. This suggests that similarities between HIV-1 and SHIV landscapes are not solely determined by potential biases in the inference approach. We have now added a reference to this point in the main text:

In contrast, the inferred fitness landscapes of CH505 and CH848, which share few mutations in common, are poorly correlated (Supplementary Fig. 6). This suggests that the similarities between viral fitness values in humans and RMs are not artifacts of the model, but rather stem from similarities in underlying evolutionary drivers.

The timing evidence is slightly weakened by the fact that bNAb detection is different from bNAb presence and the possibility that fitness increases occurred after the bNAbs appeared remains. Still, their conclusion is plausible and fits in with the other observations which form a coherent and compelling picture.

Yes, we agree that this is a limitation of our analysis — bNAbs may have been present at low levels before they were detected, and we cannot definitively reject selection by bNAbs. Nonetheless, in at least one case (RM5695), rapid fitness gains were substantially separated in time from bNAb detection (roughly 2 weeks after infection vs. 16 weeks, respectively). We have now added this point in a new paragraph in the Discussion:

While we found a strong relationship between viral fitness dynamics and the emergence of bnAbs, it may not be true that the former stimulates the latter. For example, bnAbs may have been present within each host before they were experimentally detected. Rapid viral fitness gains within hosts that developed broad antibody responses could then have been driven by undetected bnAb lineages. However, we did not find strong selection for known bnAb resistance mutations, and in at least one case (RM5695), rapid fitness gains (roughly 2 weeks after infection) substantially preceded bnAb detection (16 weeks). Still, given the limited size of the data set that we studied, it is unclear the extent to which our results will transfer to larger and broader data sets.

Overall thisrpretations could provide valuable insights into the broader significance of these results. is a convincing paper, part of a larger admirable project of accurately inferring complete fitness landscapes.
**Reviewer #3 (Public review):**
Summary:Shimagaki et al. investigate the virus-antibody coevolutionary processes that drive the development of broadly neutralizing antibodies (bnAbs). The study's primary goal is to characterize the evolutionary dynamics of HIV-1 within hosts that accompany the emergence of bnAbs, with a particular focus on inferring the landscape of selective pressures shaping viral evolution. To assess the generality of these evolutionary patterns, the study extends its analysis to rhesus macaques (RMs) infected with simianhuman immunodeficiency viruses (SHIV) incorporating HIV-1 Env proteins derived from two human individuals.Strengths:A key strength of the study is its rigorous assessment of the similarity in evolutionary trajectories between humans and macaques. This cross-species comparison is particularly compelling, as it quantitatively establishes a shared pattern of viral evolution using a sophisticated inference method. The finding that similar selective pressures operate in both species adds robustness to the study's conclusions and suggests broader biological relevance.Weaknesses:However, the study has some limitations. The most significant weakness is that the authors do not sufficiently discuss the implications of the observed similarities. While the identification of shared evolutionary patterns (e.g., Figure 5) is intriguing, the study would benefit from a more explicit discussion of what these findings mean for instance, in the context of HIV vaccine design, immunotherapy, or fundamental viral-host interactions. Even speculative inte

Thank you for this suggestion. We have now clarified the potential implications of our work in several areas. While speculative, one possible application is in vaccine design: it may be beneficial to design sequential immunogens to mimic the patterns of viral evolution associated with rapid fitness gains. This “population-based” design principle is different from typical approaches, which have focused on molecular details of virus surface proteins.

We have extended our discussion of our results in the context of viral evolution within and across hosts and related host species. Overall, our work suggests that there may be relatively few paths to significantly higher viral fitness in vivo. Evolutionary “contingencies” such as shifting immune pressure or epistatic interactions could influence the direction of evolution, but not so dramatically that the dynamics that we see in different hosts are not comparable. We have also connected our work more broadly to the literature in evolutionary parallelism in HIV-1 in different contexts.

A secondary, albeit less critical, limitation is the placement of methodological details in the Supplementary Information. While it is understandable that the authors focus on results in the main text - especially since the methodology is not novel and has been previously described in earlier publications - some readers might benefit from a more thorough presentation of the method within the main paper.

We have now modified the main text to add a new section, “Model overview,” that lays out the key steps of our approach. While we reserve technical details for the Methods, we believe that this new section provides more intuition about how our results were obtained (including a discussion of the important Eq. 12, now Eq. 3 in the main text) and our underlying assumptions.

Conclusions:Overall, the study presents a compelling analysis of HIV-1 evolution and its parallels in SHIV-infected macaques. While the quantitative comparison between species is a notable contribution, a deeper discussion of its broader implications would strengthen the paper's impact.
**Reviewer #1 (Recommendations for the authors):**
I suggest de-emphasizing bnAbs and focusing on selection landscape inference, which seems to be the actual focus of the paper.

While we do not directly study antibody development in this work, bnAb development is certainly an important motivating factor. As described in the responses above, we have now modified the Abstract and Discussion to place relatively more emphasis on fitness comparisons and to relatively less focus on bnAb development.

**Reviewer #2 (Recommendations for the authors):**
Please make sure that the MPL method is defined in this paper and its limitations are at least partially repeated.

As noted in responses above, we have now included more methodological details in the main text of the paper, which we hope will make the intuition and assumptions involved in our analysis clearer.

I'd like the code to better show or describe the model, I could not figure out the model details by looking at the code. It seems mostly just to be csv exporting for use with preexisting MPL code. A longer code readme would be helpful.

We have now updated the README on GitHub to include a conceptual overview of our inference approach, which references how each step is implemented in the code.

**Reviewer #3 (Recommendations for the authors):**
Try to give some more details (not necessarily giving the full mathematical derivation) on the statistical method utilized.

As noted above, we have now expanded our discussion of the statistical methods and assumptions in the main text.

Figures 3 and 4 are somewhat 'messy'. Although I do not have a constructive suggestion here, I feel that with a little more effort maybe the authors could come up with something more clean.

It is true that the mutation frequency dynamics are somewhat “choppy” and difficult to follow intuitively. To attempt to make these figures easier to parse visually, we have increased the transparency on the lines and added exponential smoothing to the mutation frequencies, resulting in smoother trajectories. The trajectories without smoothing are retained in Supplementary Fig. 3. Here we also note that this smoothing is for visual purposes only; we use the original frequency trajectories for inference, rather than the smoothed ones.